# CEP97 phosphorylation by Dyrk1a is critical for centriole separation during multiciliogenesis

Moonsup Lee[1], Kunio Nagashima[2], Jaeho Yoon[1], Jian Sun[1], Ziqiu Wang[2], Christina Carpenter[2], Hyun-Kyung Lee[1], Yoo-Seok Hwang[1], Christopher J. Westlake[3], and Ira O. Daar[1]

Proper cilia formation in multiciliated cells (MCCs) is necessary for appropriate embryonic development and homeostasis. Multicilia share many structural characteristics with monocilia and primary cilia, but there are still significant gaps in our understanding of the regulation of multiciliogenesis. Using the *Xenopus* embryo, we show that CEP97, which is known as a negative regulator of primary cilia formation, interacts with dual specificity tyrosine phosphorylation regulated kinase 1A (Dyrk1a) to modulate multiciliogenesis. We show that Dyrk1a phosphorylates CEP97, which in turn promotes the recruitment of Polo-like kinase 1 (Plk1), which is a critical regulator of MCC maturation that functions to enhance centriole disengagement in cooperation with the enzyme Separase. Knockdown of either CEP97 or Dyrk1a disrupts cilia formation and centriole disengagement in MCCs, but this defect is rescued by overexpression of Separase. Thus, our study reveals that Dyrk1a and CEP97 coordinate with Plk1 to promote Separase function to properly form multicilia in vertebrate MCCs.

## Introduction

Multiciliation is a developmental process that produces up to a few hundred cilia on the surface of multiciliated cells (MCCs). The coordinated beating of multicilia generates directional fluid flow in specific tissues such as adult brain ventricles, the oviduct, and airways (Brooks and Wallingford, 2014; Spassky and Meunier, 2017), as well as in epithelia of the *Xenopus* embryo (Werner and Mitchell, 2012), contributing to development and homeostasis. Multiciliation consists of sequential steps. First, a large number of centrioles are amplified through the deuterosomes, parental centrioles, and even a cloud of pericentriolar and fibrogranular material (Klos Dehring et al., 2013; Mercey et al., 2019; Zhao et al., 2019, 2013). The newly synthesized mature centrioles disengage from deuterosomes and parental centrioles through the function of cell cycle regulators such as Polo-like kinase1 (Plk1), CDC20B, and Separase (Al Jord et al., 2017; Revinski et al., 2018), and the centriole number scales with the apical area of MCCs (Kulkarni et al., 2021). Subsequently, these modified centrioles (basal bodies) migrate and dock to the apical surface of MCCs, which are dependent on an actin cytoskeleton meshwork (Boisvieux-Ulrich et al., 1990). Simultaneously, accessory structures required for ciliogenesis are assembled on the basal bodies, which is followed by cilium elongation and directional ciliary beating (Wallingford, 2010; Zhang and Mitchell, 2016).

Centrosomal protein 97 (CEP97) was originally found to suppress primary cilia formation in collaboration with CP110 by capping the mother centriole (Spektor et al., 2007), and the removal of CEP97 and CP110 from the mother centriole is a prerequisite for primary cilia formation (Huang et al., 2018; Nagai et al., 2018). However, recent studies demonstrate that CP110 is required in both primary and multiciliogenesis in vivo in different species (Walentek et al., 2016; Yadav et al., 2016), and CEP97 is reported to also be essential for cilia formation in *Drosophila* by modulating centriole stability (Dobbelaere et al., 2020). While CP110 collaborates with CEP97 to modulate ciliogenesis in cultured cells, it has been suggested that CP110 may regulate ciliogenesis independently of CEP97 in in vivo model systems (Dobbelaere et al., 2020; Galletta et al., 2016; Walentek et al., 2016). The role of CEP97 in multiciliation in vertebrates remains to be investigated.

The dual specificity tyrosine phosphorylation regulated kinase 1A (Dyrk1a) gene is located within the Down syndrome critical region of chromosome 21 and is a serine/threonine kinase that has been studied mostly in neuronal development and brain physiology (Duchon and Herault, 2016). Nevertheless, Dyrk1a also participates in a wide range of physiological functions, such as proliferation, differentiation, and apoptosis, through phosphorylating diverse substrates, depending on the cellular context (Fernández-Martínez et al., 2015). A recent report suggests that Dyrk1a activity is crucial to proper ciliogenesis in *Xenopus* MCCs, yet the molecular mechanism is still unknown (Willsey et al., 2020). Although the consensus sequence

[1]National Cancer Institute, Frederick, MD;   [2]Electron Microscopy Laboratory, Frederick National Laboratory for Cancer Research, Frederick, MD;   [3]Laboratory of Cellular and Developmental Signaling, Center for Cancer Research, National Cancer Institute, National Institutes of Health, Frederick, MD.

Correspondence to Ira O. Daar: daari@mail.nih.gov.

recognized by Dyrk1a consists of serine/threonine followed by proline at position +1 and arginine at position −2 or −3 relative to serine/threonine, the recognition sequences vary considerably among substrates (Soundararajan et al., 2013).

Here, we identify CEP97 as a substrate of Dyrk1a that is required for cilia formation in vertebrate MCCs. Intriguingly, CEP97 phosphorylation status appears to be important for proper basal body behavior and multiciliation. We provide evidence that an interaction between CEP97 and Dyrk1a leads to phosphorylation of CEP97, which induces the recruitment of Plk1 to promote the proper formation of basal bodies. This concept is supported by the rescue of impaired multiciliation in CEP97 and Dyrk1a morphant MCCs by overexpression of Separase (a downstream effector of Plk1). The Plk1-Separase signaling pathway contributes to the process of centriole disengagement during MCC maturation, and our study reveals a signaling axis of Dyrk1a–CEP97–Plk1 that contributes to disengagement of mature centrioles and ciliogenesis in vertebrate MCCs.

## Results

### CEP97 associates with Dyrk1a
To study the role of CEP97 in vertebrate multiciliogenesis, we first identified novel binding partners of CEP97 by performing immunoprecipitation (IP) coupled to mass spectrometry (MS) using neurula stage *Xenopus* embryos expressing exogenous CEP97 (Table S1). Dyrk1a was obtained from two independent MS experiments. To validate the interaction between CEP97 and Dyrk1a, co-IP experiments from embryos exogenously expressing tagged versions of both proteins were performed. The exogenous V5-tagged CEP97 was detected in Flag-tagged Dyrk1a immune complexes, and in reciprocal co-IPs, Flag-tagged Dyrk1a was found in association with V5-tagged CEP97 (Fig. 1 A). An endogenous protein interaction was also confirmed by IP of CEP97 from HEK293T cells in which Dyrk1a was detected in the immune complexes (Fig. 1 B).

### The aa regions (541–565) of CEP97 and the histidine (His) repeat of Dyrk1a are required for CEP97–Dyrk1a association
To identify the regions within each protein critical for the CEP97–Dyrk1a association, we conducted protein interaction domain mapping. Internal deletion mutants of CEP97 were generated based on the location of the isoleucine and glutamine (IQ) motif (518–537) of CEP97 because the IQ motif plays a known role in protein–protein interactions (Campiglio et al., 2018; Li and Sacks, 2003; Tidow and Nissen, 2013). Co-IP assays showed that the Δ516–540 aa mutant lacking the IQ motif was still able to interact with Dyrk1a; yet the deletion of the region adjacent to the IQ motif (aa 541–565) caused a significant reduction in CEP97–Dyrk1a complex formation (Fig. 1, C and D). Conversely, various serial Dyrk1a deletion mutants were used to map domains responsible for an interaction with CEP97. Reciprocal co-IPs revealed that a Dyrk1a mutant lacking the His repeat most effectively lost the interaction with CEP97 in both directions of co-IPs, although a noticeable variable decrease in ΔC mutant–CEP97 binding was also shown, but only in one direction of co-IP (CEP97-HA immune complexes; Fig. 1, E and F).

To assess whether the kinase activity of Dyrk1a was important for the Dyrk1a–CEP97 interaction, a kinase-dead form (K180R) of *Xenopus laevis* Dyrk1a was tested in the co-IP. The loss of kinase activity did not diminish the interaction with CEP97 (Fig. 1 G), suggesting that the enzymatic activity of Dyrk1a does not affect the CEP97–Dyrk1a interaction.

### CEP97 is a novel substrate of Dyrk1a
A noticeable mobility shift of CEP97 protein on immunoblots implicated a potential post-translational modification such as phosphorylation (Fig. 1 H). To assess whether CEP97 was phosphorylated, the embryonic lysates expressing exogenous CEP97 were incubated with λ-phosphatase (λ PPase) in the absence of protein phosphatase inhibitors. The λ PPase incubation completely removed the slower mobility band in a manner similar to Dishevelled 2 (Dvl2), the positive control in this assay (Fig. 1 H). To determine whether CEP97 might be a substrate for Dyrk1a-mediated phosphorylation, a recombinant CEP97 protein was incubated with active Dyrk1a in vitro. Of note, CEP97 migrated slower in the gel in the presence of active Dyrk1a, while this mobility shift was reversed by λ PPase treatment (Fig. 1 I). As expected, active Dyrk1a did not affect Dvl2 (a nonsubstrate control) migration in the gel (Fig. 1 I). To identify the putative phosphorylation residues of CEP97, the purified recombinant CEP97 was incubated with active Dyrk1a and mass spectrometric analysis was performed. This analysis revealed that Dyrk1a phosphorylated CEP97 in vitro on serines 634, 643, 649, and 653 located in a region conserved among different species (Fig. 1 J and Table S2).

### CEP97 phosphorylation is required for ciliogenesis in MCCs
Although CEP97 in collaboration with CP110 suppresses primary cilia formation in cultured cells (Huang et al., 2018; Nagai et al., 2018; Spektor et al., 2007) and regulates centriole length in *Drosophila* (Galletta et al., 2016), a recent study demonstrated that loss of CEP97 compromises cilia formation in *Drosophila* (Dobbelaere et al., 2020). To test whether CEP97 is required for ciliogenesis in vertebrates, we conducted knockdown experiments using morpholino oligonucleotides (MOs) against CEP97, and rescue experiments were performed using MO-resistant CEP97 mRNA (Fig. S1, A and B). Unlike control morphants, CEP97 morphants showed a noticeable reduction of acetylated tubulin staining (a cilia marker) in the MCCs (Fig. 2, A and B). The CEP97 knockdown-mediated defects in multiciliation were phenocopied by genetic disruption of CEP97 using Crispr/Cas9 knockout technology (Fig. S1, C and D). Of interest, while the reexpression of WT CEP97 or a phosphomimetic mutant (SD; S634D, S643D, S649D, and S653D) with an equivalent protein expression level led to a marked rescue of the acetylated tubulin staining in CEP97 morphant MCCs, a similar amount of CEP97 phospho-null mutant (SA; S634A, S643A, S649A, and S653A) failed to restore the acetylated tubulin staining (Fig. 2, A–C). These data indicate that phosphorylation of CEP97 on the identified Dyrk1a sites plays a critical role in multiciliation. As a further test of whether an interaction between CEP97 and Dyrk1a is also required for ciliation, the CEP97 mutant (lacking aa 541–565) that is unable to associate with Dyrk1a was

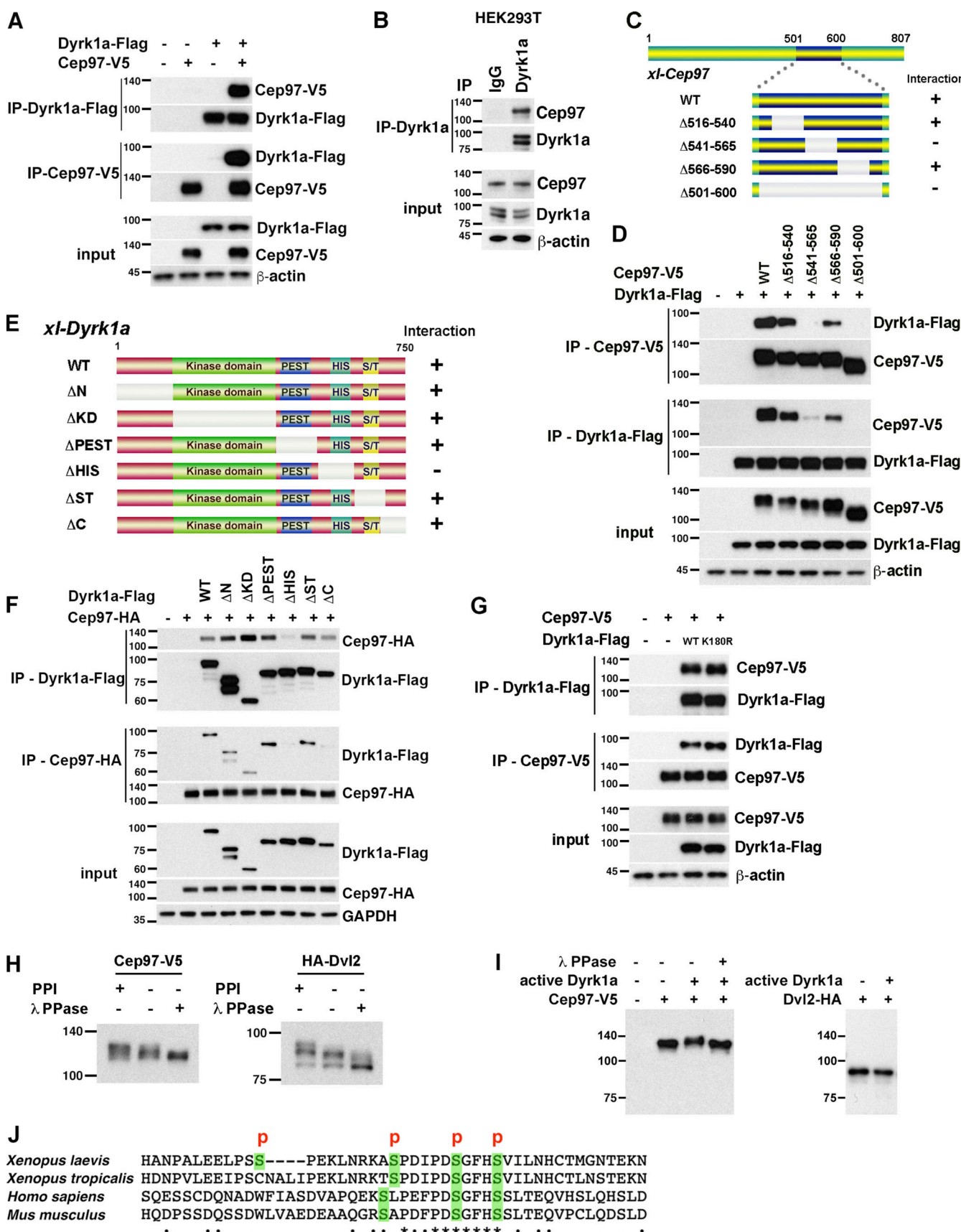

Figure 1. **CEP97 associates with Dyrk1a. (A)** Exogenous CEP97 interacts with Dyrk1a. The indicated mRNAs were injected into *Xenopus* embryos, and co-IPs were conducted. **(B)** Endogenous CEP97 interacts with Dyrk1a. Endogenous Dyrk1a was precipitated from HEK293T cell lysates and immunoblotted with the

indicated antibodies. **(C)** Schematic diagram of CEP97 deletion mutants and depiction of ability in co-IP with Dyrk1a. **(D)** The aa region 541–565 of CEP97 is required for Dyrk1 interaction. Using embryos expressing the indicated proteins, reciprocal co-IPs were performed. **(E)** Schematic diagram of Dyrk1a deletion mutants and depiction of ability in co-IP with CEP97. Representative domains are indicated: WT; N, N-terminus; KD, kinase domain; PEST, domain rich in proline, glutamate, serine, and threonine; His, histidine repeat; ST, serine/threonine-rich region; C, C-terminus. **(F)** His region of Dyrk1a is required for CEP97 interaction. Embryo lysates expressing the indicated proteins were used for reciprocal co-IP assay. **(G)** Dyrk1a kinase activity is not necessary for CEP97– Dyrk1a interaction. The indicated mRNAs were injected into embryos, and lysates were used for co-IPs. **(H)** Embryo lysates expressing the indicated protein were incubated with or without λ PPase at 30°C for 30 min and analyzed in 6% polyacrylamide gel. PPI, phosphatase inhibitor. **(I)** Dyrk1a-mediated CEP97 protein shift was reversed by λ PPase. The indicated recombinant proteins were incubated with active Dyrk1a and/or λ PPase at 30°C for 30 min. **(J)** CEP97 amino acid sequence comparison among different species. The region including the residues that Dyrk1a phosphorylates is conserved among different species. P, phosphorylation; conservation: fully conserved (*), strongly conserved (:), weakly conserved (.). Numbers marked on left side of Western blot images represent protein molecular weight (kilodaltons).

introduced into CEP97 morphants. Strikingly, this interaction mutant did not restore the acetylated tubulin staining in CEP97 morphants (Fig. S2, A and B). Thus, our findings suggest that Dyrk1a-mediated phosphorylation of CEP97 plays a critical role in multiciliation.

Having established the requirement for CEP97 for proper ciliogenesis as measured by acetylated tubulin staining, we performed knockdown experiments targeting Dyrk1a. We found that loss of Dyrk1a also decreased multiciliation, as evidenced by acetylated tubulin in MCCs (Fig. 2, D and E; and Fig. S2, C and D), which is consistent with a previous report (Willsey et al., 2020). While multiciliation in Dyrk1a morphants was rescued by re-introduction of WT Dyrk1a, expression of the kinase-dead Dyrk1a mutant (K180R) failed to restore proper ciliogenesis in MCCs (Fig. S2, C and D). The percentage of MCCs per imaged area in CEP97 and Dyrk1a morphants was not altered when compared with control morphant MCCs (Fig. S2, E and F).

To further test the requirement of the Dyrk1a–CEP97 interaction for proper multiciliation, a Dyrk1a mutant lacking the His domain required for the association with CEP97 was expressed in Dyrk1a morphants. The Dyrk1a mutant failed to rescue the Dyrk1a knockdown phenotype (Fig. 2, D and E), indicating a critical role for the CEP97–Dyrk1a interaction in multiciliation. Using scanning EM (SEM), we observed a marked decrease in the population and length of multicilia in the CEP97 and Dyrk1a morphant MCCs (Fig. 2, F and G), confirming that CEP97 and Dyrk1 are required for multiciliation in MCCs. Thin-section transmission EM (TEM) of the remaining axonemes of MCCs in CEP97 and Dyrk1a morphants did not reveal any incomplete or missing axoneme structures, as had been observed in the spermatids of CEP97 mutant flies (Dobbelaere et al., 2020; Fig. S2 G).

To address whether CEP97 and Dyrk1a might play the same role in motile monocilia as we demonstrated for multicilia, we examined the gastrocoel roof plate (GRP) in embryos injected with the CEP97 or Dyrk1a MOs. Compared with control morphants, the length of GRP cilia in CEP97 morphants was reduced to ~50% of controls (Fig. S2, H and I). Expression of either the phospho-null or phosphomimetic mutants of CEP97 was sufficient to restore the CEP97 knockdown phenotypes in the GRP (Fig. S2, H and I). In contrast, Dyrk1a morphants did not display a decrease in length of the GRP cilia (Fig. S2, H and I). These data suggest that CEP97 may regulate ciliogenesis in the GRP independent of Dyrk1a phosphorylation.

As a prerequisite for proper ciliogenesis, de novo centrioles/ basal bodies should migrate and dock with the apical surface of MCCs (Brooks and Wallingford, 2014). The migration and docking of basal bodies depend on the formation of the actin meshwork (Antoniades et al., 2014; Kulkarni et al., 2018; Werner et al., 2011). Using TEM, we observed impaired migration and docking of basal bodies in the MCCs of CEP97 and Dyrk1a morphant epidermis (Fig. 2 F). The abnormal migration and docking phenotypes were confirmed by immunofluorescence using centrin4-RFP (Cent-RFP) as a basal body marker (Fig. 3 A). In control MCCs, Cent-RFP localized to the apical surface (marked by phalloidin) of MCCs; however, Cent-RFP did not properly locate to the apical surface of CEP97 and Dyrk1a morphant MCCs (Fig. 3 A). As expected, when the apical actin meshwork is disrupted, the phalloidin staining declined in the CEP97 and Dyrk1a morphant MCCs (Fig. 3, A–C). To a significant extent, the failure of basal body migration and docking, along with the loss of apical actin, was restored by WT CEP97 expression, but not by the phospho-null mutant (Fig. 3, A–C).

Because proper distal appendage formation is required for basal body docking and subsequent ciliogenesis (Siller et al., 2017), we assessed whether the basal bodies in CEP97 and Dyrk1a morphant MCCs contain distal appendage structures using anti-Cep164 antibody. 3D structured illumination microscopy (3D-SIM) analysis showed that endogenous Cep164 decorated Cent-RFP in all experimental groups including CEP97 and Dyrk1 morphant MCCs, suggesting that loss of CEP97 and Dyrk1a does not affect distal appendage assembly. Cep164 localization in the control morphants and CEP97 morphants rescued by WT CEP97 appears somewhat different from that in CEP97 and Dyrk1a morphants (Fig. 3 D). In the Dyrk1a and CEP97 morphants, Cep164 seems to be recruited to a portion of the centriole circumference but does not extend fully around it, suggesting a possible effect on centriole maturation. Interestingly, the loss of CEP97 and Dyrk1a caused a severe spacing defect between basal bodies, which was not rescued by CEP97 phosphomutant (SA) expression (Fig. 3 D). Taken together, the cooperation of CEP97 and Dyrk1a is required for a proper basal body docking to the apical surface and multiciliation.

## CEP97 colocalizes with Plk1 in migrating MCCs

Although CEP97 is known to localize to the centrioles of primary cilia in mammalian cultured cells (Korzeniewski et al., 2010; Spektor et al., 2007), the localization pattern of CEP97 in *Xenopus* MCCs is somewhat different from that in cultured cells (Walentek et al., 2016). Immunofluorescence assays using a confocal microscope showed that CEP97-GFP localized near the

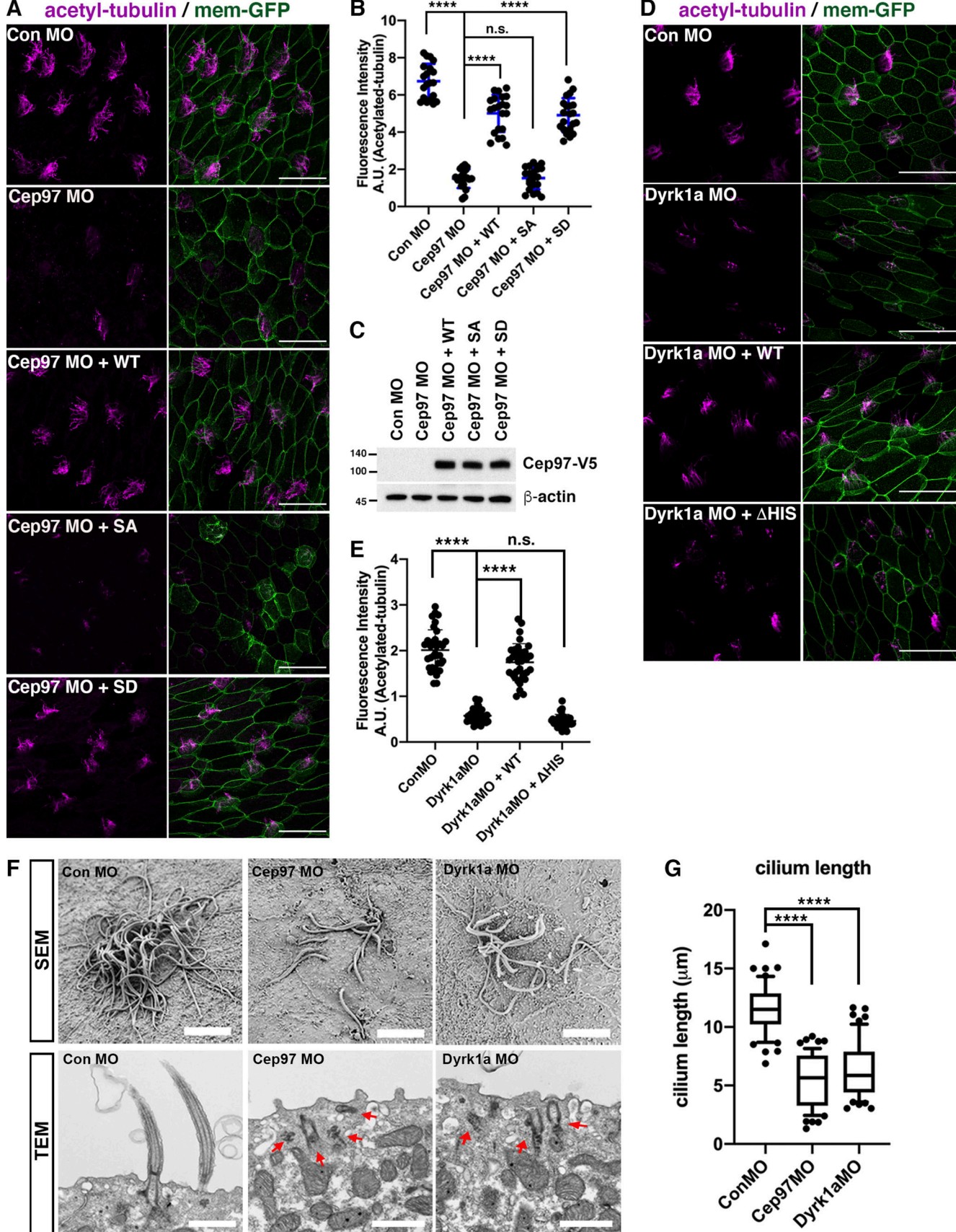

Figure 2.   **CEP97-Dyrk1a modulates multiciliation in *Xenopus* MCCs. (A and D)** The acetylated tubulin signal in MCCs is decreased upon CEP97 knockdown (A) and Dyrk1 knockdown (D). A cocktail of mRNAs and morpholinos was injected into one ventral blastomere in eight-cell stage embryos, cultured to stage 27,

and fixed. Multicilia were visualized by antiacetylated tubulin staining (magenta), and membrane GFP (green) was used as a tracer. Scale bars, 50 µm. Con, control. **(B and E)** Quantification of relative acetylated tubulin intensity in A (number of images for quantification, n = 40; embryos per group from three independent experiments for each condition, n = 20) and D (number of images for quantification, n = 32; embryos per group from three independent experiments for each condition, n = 16). ****, P < 0.0001; one-way ANOVA. Error bars indicate ±SD. **(C)** Immunoblot of the indicated exogenous proteins expressed for the rescue experiment in A. Numbers marked on left side of Western blot images represent protein molecular weight (kilodaltons). **(F)** SEM and TEM of embryonic epidermis at stage 26. SEM revealed decreased number and length of multicilia in CEP97 and Dyrk1a morphant MCCs. The indicated MOs were injected into both ventral blastomeres at the eight-cell stage. Red arrows point to mispositioned basal bodies. Scale bars, 5 µm (SEM) and 1 µm (TEM). **(G)** Quantification of cilia length per MCC, n = 10; embryo per group, n = 5; ****, P < 0.0001, unpaired two-tailed t test. Error bars indicate ±SD.

basal bodies (Cent-RFP) in MCCs at developmental stage 23 to some extent; however, CEP97-GFP was rather broadly distributed over the surface of MCCs (Fig. S3 A), which is consistent with a previous report (Walentek et al., 2016). Dyrk1a-GFP showed clear localization at or near Cent-RFP in the basal bodies at this stage (Fig. S3 B). In contrast, the immature migrating MCCs (stage 19) displayed CEP97-GFP localization near or at the centrioles (Cent-RFP; Fig. S3 C), suggesting a potential role of CEP97 in the centriole production and/or maturation in MCC progenitors.

Plk1 performs a variety of essential roles in mitotic progression, including centriole maturation and disengagement (Kim et al., 2015; Lončarek et al., 2010). In the context of ciliogenesis, Plk1 stimulates primary cilia disassembly in cultured cells (Lee et al., 2012; Zhang et al., 2017), while in MCCs, Plk1 activity is required for centriole disengagement (Al Jord et al., 2017; Revinski et al., 2018). Our IP-MS data provided a strong indication that Plk1 also associates with CEP97 (Table S1); thus, we examined whether CEP97 and Plk1 are both found in centrioles of migrating MCCs. Using immunofluorescence, we validated the localization of GFP-tagged Plk1 and CEP97 to the centrioles (Cent-RFP) just below the apical surface in migrating MCCs (Fig. S3 D). We then tested whether Plk1 and CEP97 proteins are colocalized at the same developmental stage. The MCCs coexpressing CEP97-mCherry and Plk1-GFP displayed a significant degree of colocalization (Fig. 4 A) of these proteins, indicating that both proteins are present at the centrioles of migrating MCCs.

Because the Polo-box (PB) domain of Plk1 recognizes phospho-threonine/serine residues for associating with binding partners (Zitouni et al., 2014), CEP97-Plk1 binding was tested using the phospho-null (SA) and phosphomimetic (SD) mutants of CEP97. The Dyrk1a phosphorylation site in Cep97 does not resemble the core consensus motif for Plk1 docking (S-[pT/pS]-[P/X]; X being any amino acid except Cys; Elia et al., 2003); yet Plk1 can associate with its binding partners in a noncanonical manner (Archambault et al., 2008; Bonner et al., 2013). Co-IP analysis indicated that the CEP97 SA mutant was markedly impaired in the ability to interact with Plk1, but the WT CEP97 and the CEP97 SD mutant sustained the ability to bind Plk1 (Fig. 4 B). Of note, the CEP97 mutant lacking aa 541–565 did not interact with Plk1 (Fig. 4 C), suggesting that CEP97 association with Dyrk1a may be critical for CEP97-Plk1 binding. In addition, we employed Plk1 constructs that lack the PB1 and PB2 domains (ΔPB) or that consist of the two PB domains alone (PB) to test whether the PB domain is required for CEP97-Plk1 binding. As expected, the mutant lacking the PB domains failed to effectively bind CEP97, while the PB domains alone were sufficient to

associate with CEP97 (Fig. 4, D and E). The CEP97 SA mutant was unable to associate with either WT Plk1 or the PB domains alone (Fig. 4 F), confirming that phosphorylation of CEP97 is important for CEP97-Plk1 binding.

Plk1 was proposed to be part of a complex that forms with Cdc20B and sperm-associated antigen 5 (Spag5) to regulate centriole disengagement (Revinski et al., 2018). Although we did not identify Cdc20B or Spag5 in our IP-MS analysis in embryos, we tested this concept. Human forms of Cdc20B-Flag and Spag5-Flag were coexpressed with human CEP97-HA (WT or Δ577–610; human CEP97 deletion mutant equivalent to Xenopus CEP97 Δ541–565) in HEK293T cells, and co-IP assays were performed. While both hCdc20B-Flag and hSpag5-Flag associated with WT hCEP97-HA, the Δ577–610 mutant CEP97-HA failed to interact with hCdc20B-Flag and hSpag5-Flag (Fig. S3, E–G), suggesting that CEP97 may complex with Plk1-Cdc20B-Spag5 in the regulation of centriole disengagement in MCCs.

Because our data indicated that CEP97 associates with both Dyrk1a and Plk1, we addressed whether CEP97 may act as a scaffold between Dyrk1 and Plk1. Initially, we tested whether exogenously expressed Dyrk1a could be detected in immune complexes of exogenously expressed Plk1 and vice versa. Of note, each protein was detected in the immune complex of the other protein (Fig. 4 G). We then tested whether loss of endogenous CEP97 affected the ability of Dyrk1a to interact with Plk1. Co-IP analysis was performed on lysates from embryos injected with the mRNAs of Dyrk1a-HA and Plk1-Flag and MOs against CEP97. While CEP97 knockdown diminished the interaction between Dyrk1a-HA and Plk1-Flag, introduction of a morpholino-resistant CEP97 mRNA restored the Plk1–Dyrk1 interaction (Fig. 4, G and H). These data implicated CEP97 as a critical link in the association between Dyrk1a and Plk1. As a further test of this link, we coexpressed in embryos a WT Plk1 with a Dyrk1a mutant lacking the His region, which is critical for the association with CEP97 (Fig. 1 F), and we performed reciprocal co-IPs. The Dyrk1 ΔHis mutant did not form a complex with Plk1 (Fig. 4 I), confirming that the association between Dyrk1a and Plk1 requires CEP97.

## CEP97 affects centriole disengagement in MCCs

Plk1 is a major regulator of centriole disengagement (Kim et al., 2015; Revinski et al., 2018), and Plk1 localizes to deuterosomes (Revinski et al., 2018). To gain more precise CEP97 localization information in MCCs during centriole disengagement, we performed immunofluorescence assays with SIM using N-terminal tagged Ccdc78 as a deuterosome marker (Klos Dehring et al., 2013). SIM analysis revealed a strong association of CEP97-HA with GFP-xCcdc78 in the MCCs migrating to the apical surface,

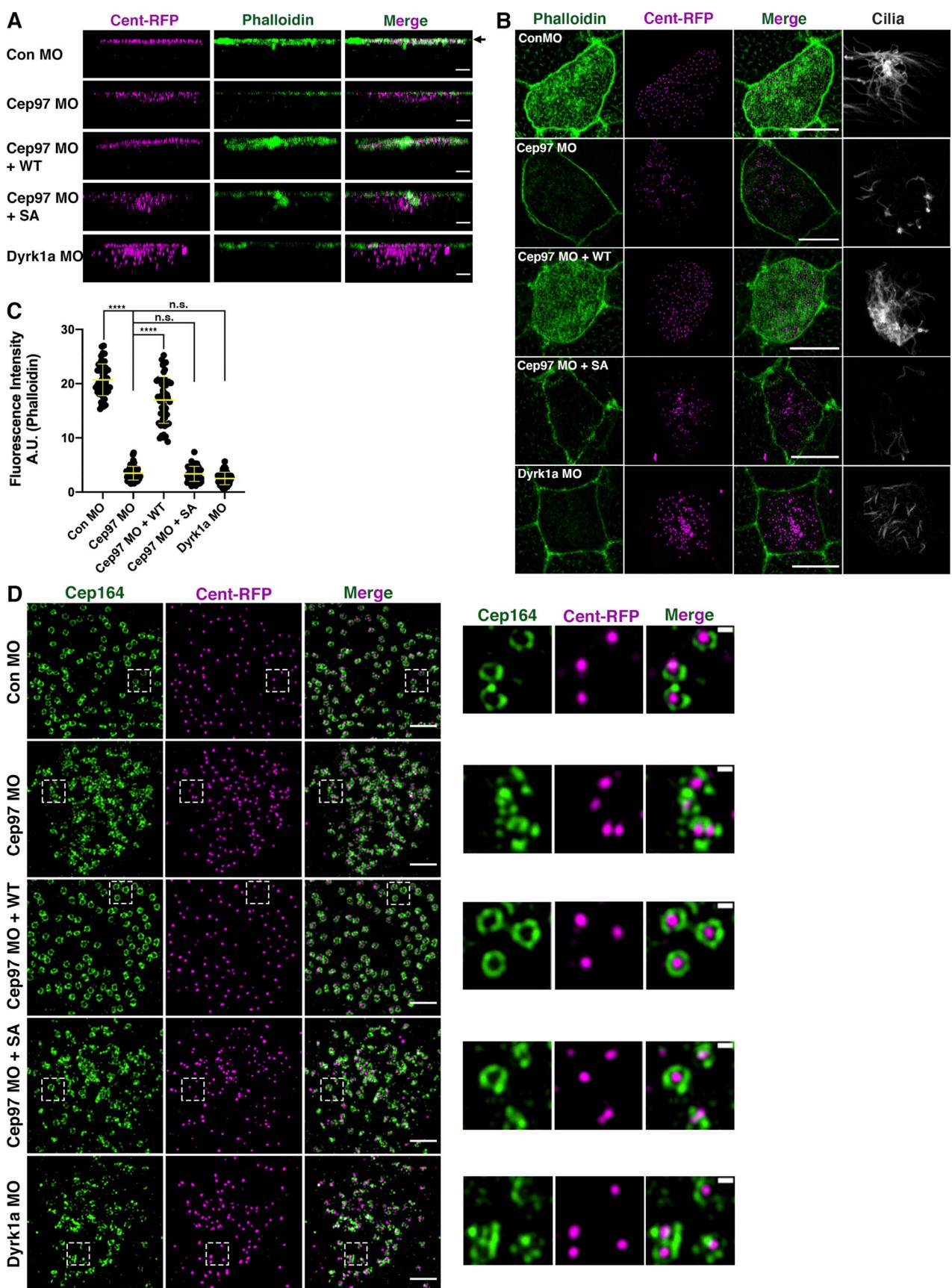

Figure 3. **Abnormal basal body migration and docking in CEP97 and Dyrk1a morphant MCCs. (A)** Defective basal body migration and docking to the apical surface of MCCs upon knockdown of CEP97 and Dyrk1a. The injected embryos were fixed at stage 25 and stained with phalloidin (green) to visualize

apical F-actin. Cent-RFP (magenta) marks basal bodies. Arrow marks apical surface. Serial z-stack confocal images were projected in the y–z plane. Scale bars, 3 μm. **(B)** Apical actin meshwork decreases in CEP97 and Dyrk1a morphant MCCs. Embryos were fixed at stage 25. Images were generated by maximum-intensity projection of serial confocal z-stack images from surface (0 μm) to subapical (−2.5 μm). Cent-RFP (magenta) marks basal bodies. Phalloidin (green) stains apical F-actin. Antiacetylated tubulin antibody stains cilia. Scale bars, 10 μm. **(C)** Quantification of phalloidin intensity in an MCC in B. ****, P < 0.0001, one-way ANOVA; MCCs, n = 45; embryos per group from one representative of three independent experiments, n = 9. Data are mean ± SD. **(D)** Basal bodies contain distal appendages in CEP97 and Dyrk1a morphant MCCs. Embryos injected with the indicated mRNAs and MOs were harvested at stage 27. Images were generated using 3D-SIM. Images in white dotted squares are magnified to the right. Data are representative of three independent experiments. Scale bars, large images = 2 μm; small images = 0.3 μm.

and the association between CEP97-HA and centrin mostly occurred at deuterosomes (GFP-xCcdc78). In contrast, CEP97-HA scarcely colocalized with centrioles separated from deuterosomes (Fig. 5 A). Interestingly, MO-mediated loss of Dyrk1a did not alter CEP97-HA localization to deuterosomes, and loss of either CEP97 or Dyrk1a did not significantly change the Plk1-HA association with deuterosomes (GFP-Ccdc78; Fig. S4, A and B). These data suggest that although Dyrk1a phosphorylation of CEP97 is important for an interaction with Plk1, recruitment of CEP97 and its binding partner Plk1 to the centriole–deuterosome complex is independent of this event.

CEP97 localization to the centriole–deuterosome complexes begged the question whether CEP97 plays a potential role in the deuterosomes during centriole maturation and disengagement. To test this hypothesis, CEP97 or Dyrk1a MOs were coinjected with mRNAs of Cent-RFP (basal body/centriole marker) and GFP-xCcdc78 or xDeup1-GFP (deuterosome markers), and centriole disengagement was examined. Control morphant MCCs displayed >90% of centriole disengagement (Fig. 5, B and C), which is in line with a previous report (Revinski et al., 2018). However, >75% of CEP97 morphant MCCs and >85% of Dyrk1a morphant MCCs failed to complete centriole disengagement (Fig. 5, B and C). Although the percentage of compromised centriole disengagement decreased at stage 25 and stage 27 to some extent, a markedly significant percentage of CEP97 and Dyrk1a morphant MCCs at these stages still contained incomplete separation of Deup1-GFP and Cent-RFP. (Fig. S5, A–D). The centriole disengagement defect caused by the CEP97 knockdown was substantially rescued by WT CEP97 expression, but the CEP97 SA mutant failed to significantly reverse the phenotype (Fig. 5, B and C).

From our data, a working model emerges suggesting that Dyrk1a associates with CEP97 and phosphorylates it. Upon phosphorylation, CEP97 is able to bind Plk1, which in turn regulates Separase activity that is responsible for the disengagement of centrioles from deuterosomes. The separation of centrioles from deuterosomes is a prerequisite for proper maturation, migration, and docking of basal bodies during multiciliation. To test this concept, we first investigated whether Separase, a key downstream target of Plk1, is required for centriole disengagement in Xenopus MCCs. Separase morphant embryos exhibited centriole disengagement defects in ~70% of MCCs at developmental stage 23, which was rescued by WT Separase expression but not by a catalytically inactive mutant of Separase (Fig. 6, A, C, and D). Of note, expression of the mutant Separase at levels equivalent to the WT Separase caused centriole disengagement defects (Fig. 6, B, E, and F). Collectively, these data indicate that Separase plays a substantive role in centriole disengagement in Xenopus MCCs.

An important functional test of our proposed model is whether Separase can rescue the impaired centriole disengagement in the CEP97 and Dyrk1a morphant MCCs. Separase expression in CEP97 or Dyrk1a morphant MCCs rescued the defects in apical actin formation and multiciliation (Fig. 7, A–C), which was confirmed by SEM clearly showing that the reduced cilia population and length were partially rescued by WT Separase expression (Fig. 7, D and E). Furthermore, the anomalous persistent deuterosome–centriole engagement resulting from the knockdown of CEP97 or Dyrk1a was also rectified by WT Separase expression (Fig. 8, A and B). However, expression of an inactive mutant Separase (at levels that do not induce any defects) failed to rescue the phenotypes in both CEP97 and Dyrk1a morphant MCCs (Fig. 8, A and B). Collectively, these experiments strongly suggest that a Dyrk1a-CEP97-Plk1-Separase signaling pathway regulates centriole disengagement in MCCs (Fig. 8 C).

## Discussion

The widely accepted role for CEP97 is to functionally collaborate with its partner CP110 in suppressing primary cilia assembly (Nagai et al., 2018; Spektor et al., 2007). Our study provides a new role for CEP97 as a scaffold for two associated kinases, Dyrk1a and Plk1, to assist in the separation, migration, and docking of basal bodies in multiciliogenesis. A recent study demonstrated that CEP97 promotes ciliogenesis in monocilia of Drosophila (Dobbelaere et al., 2020), but the mechanism of action is quite different from the role we reveal for multicilia in the epidermis. In the fly study, the supported model was one in which CEP97 functions as part of a protective cap that coordinates with the microtubule acetylation machinery to sustain the stability of centrioles for proper functioning during ciliogenesis (Dobbelaere et al., 2020). CEP97 mutant flies demonstrate incomplete or missing sperm axonemes in testis and abnormal basal bodies in mechanosensory cilia of the chordotonal organ (Dobbelaere et al., 2020). In the multicilia of CEP97 and Dyrk1a morphant frog embryos, we observed a marked reduction in cilia number and length (Fig. 2, F and G), but no defects were observed in the axoneme structure of the remaining cilia (Fig. S2 G). Because we failed to obtain ultrastructural images of multicilia basal bodies, we cannot exclude the possibility that structural abnormalities in the basal bodies of CEP97 and Dyrk1a knockdown MCCs may be present.

We observed that the knockdown of CEP97 caused ciliogenesis defects not only in MCCs (multicilia; Fig. 2) but also in the monocilia of the GRP (Fig. S2 H). Unlike multicilia, the phosphorylation status of CEP97 is dispensable for monocilia

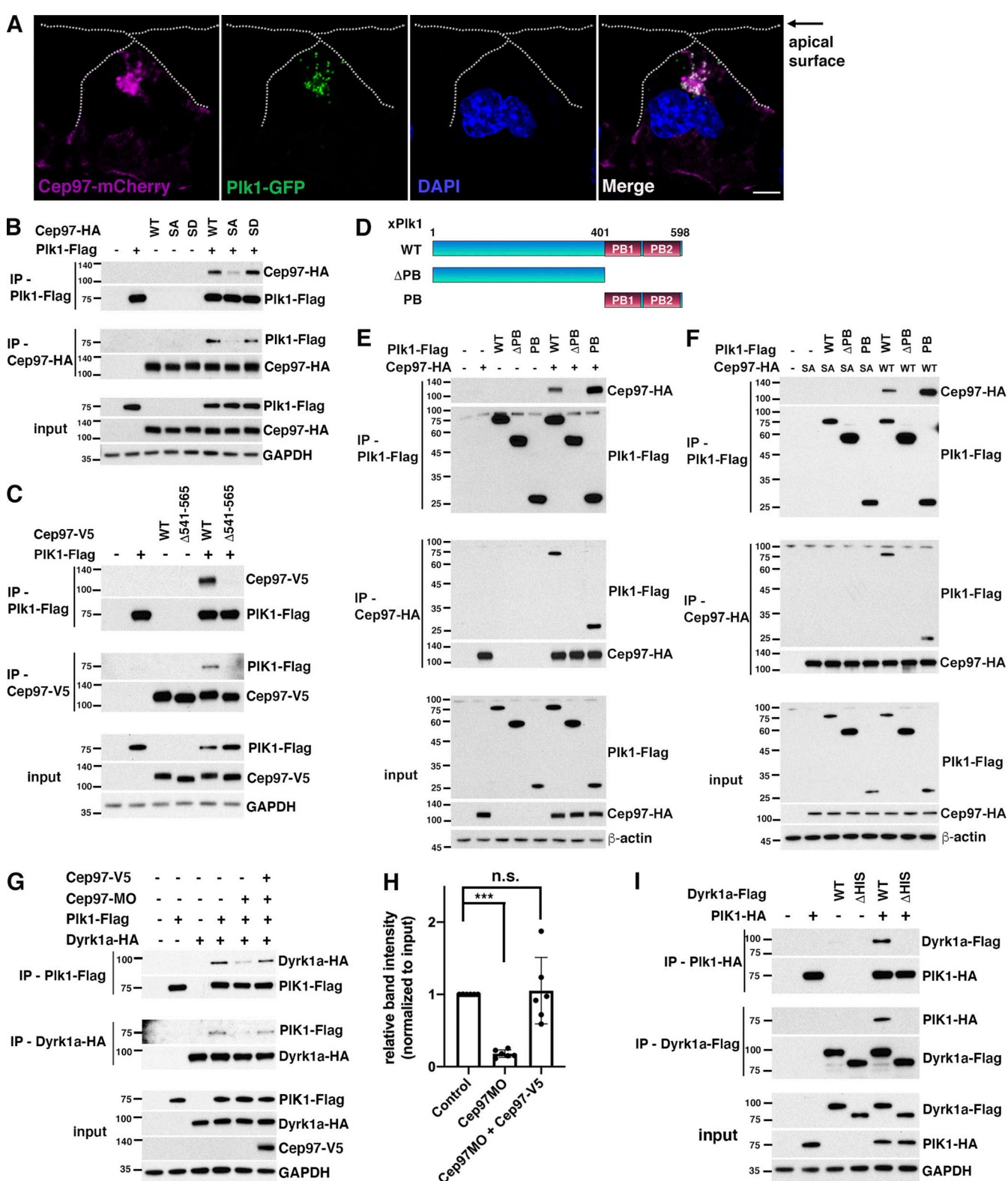

**Figure 4.** **CEP97 forms a complex with Dyrk1a and Plk1. (A)** The subcellular colocalization of CEP97 and Plk1 in migrating MCCs. Embryos injected with the indicated mRNAs were fixed at stage 19 and transversely sectioned. CEP97-mCherry, magenta; Plk1-GFP, green. DAPI was used to stain nuclei. Scale bars, 5 µm. **(B)** The phosphorylation of CEP97 is required for CEP97-Plk1 interaction. The embryos were injected with the indicated mRNAs, and reciprocal co-IPs were performed. **(C)** CEP97 mutant lacking aa 541–565 does not associate with Plk1. Reciprocal co-IPs were performed with embryo lysates as indicated. **(D)** Schematic diagram of *Xenopus* Plk1 deletion mutants. **(E and F)** PB domain is required for CEP97-Plk1 binding. CEP97 WT (E), but not CEP97 SA (F), associates with WT Plk1 and PB alone. The indicated mRNAs were injected into one-cell stage embryos, and reciprocal co-IPs were performed. **(G)** CEP97 mediates Dyrk1a-Plk1 interaction. Embryos injected with the indicated MOs or mRNA were lysed at stage 16 for co-IPs. **(H)** Quantification of co-IP protein

bands in G. Co-IP protein band density was normalized to input. Graph was generated from three independent reciprocal co-IPs. ***, P < 0.001; one-way ANOVA. Data are mean ± SD. **(I)** The His region of Dyrk1a is necessary for Dyrk1a–Plk1 interaction. The embryo lysates expressing the indicated exogenous proteins were used to perform co-IPs. Numbers marked on left side of Western blot images represent protein molecular weight (kilodaltons).

formation in the GRP, and knockdown of Dyrk1a does not affect ciliogenesis in the GRP (Fig. S2 H). Thus, even in *Xenopus*, there is a context-dependent difference in the role of CEP97 and Dyrk1a, indicating that the collaboration between CEP97 and Dyrk1a may be limited to ciliogenesis in MCCs. The collaborative role of CEP97-Dyrk1a in multiciliation is supported by two findings. First, the expression of a CEP97 phosphomutant (SA) and the Δ541–565 deletion mutant (lacking the region required for Dyrk1a binding) failed to rescue the ciliary defects in MCCs of the CEP97 morphants (Figs. 2 and S2 A). Second, the mutant Dyrk1a lacking the His region that is necessary for CEP97 binding also failed to rescue the ciliogenesis defects in MCCs of Dyrk1a morphants (Fig. 2 D).

The results from a number of experiments provide evidence to support a model (Fig. 8 C) for the role of CEP97 in multiciliogenesis: (1) Tagged CEP97 localizes to the deuterosome–centriole complex in maturing MCCs (Fig. 4 A); (2) Dyrk1a-mediated phosphorylation of CEP97 enhances Plk1 association with the complex (Fig. 4, B and F); (3) Dyrk1a-mediated phosphorylation of CEP97 is required for proper disengagement of deuterosomes (as evidenced by Ccdc78 and Deup1) from the centriole component centrin (Figs. 5 B and S5 A); and (4) overexpression of Separase in both CEP97 and Dyrk1a morphants rescues the multiciliation defects (Fig. 7), as well as the disengagement of centrioles from deuterosomes (Fig. 8 A). Thus, a model can be proposed where CEP97 interacts with and is phosphorylated by Dyrk1a, which in turn enhances Plk1 association with the CEP97–Dyrk1a complex, leading to induction of Separase-dependent centriole disengagement (Fig. 8 C).

Although CEP97 and Dyrk1a are dispensable for Plk1 localization to deuterosomes in MCCs (Fig. S4 B), the biochemistry and functional assays in vivo indicate that the interactions between these molecules are regulated by Dyrk1a phosphorylation (Figs. 1, 2, and 3). Although we did not identify peptides from CDC20B or Spag5 in the IP-MS analysis of the CEP97 immune complexes from embryos, we show that CEP97 can form a complex with Plk1, Cdc20B, and Spag5 (Fig. 4 and Fig. S3, F and G), and Spag5 has been shown to be critical for timely Separase activation (Chiu et al., 2014; Thein et al., 2007). One intriguing possibility that warrants further study is whether Dyrk1a-CEP97 may fine tune Separase activity during centriole disengagement by regulating access of Plk1-Cdc20B-Spag5 to Separase or its regulatory proteins (e.g., securin) in the centriole–deuterosome complex rather than by directly controlling Plk1 localization to the deuterosome.

Although it is clear that multiciliation defects result from the knockdown of CEP97 and Dyrk1a, we cannot exclude the possibility that these defects might be due, at least in part, to dysregulated centriole amplification. Because it is known that a massive number of centrioles are generated through the pericentriolar material (PCM) complex in deuterosomes (Boutin and Kodjabachian, 2019) and/or in a PCM cloud and fibrogranular

material in MCCs (Mercey et al., 2019), and also that Plk1 controls PCM recruitment (Kong et al., 2014), the potential impact of the collaboration of CEP97 and Dyrk1a with Plk1 on centriole amplification in MCCs needs to be investigated in the future. Moreover, because CEP97 is present early in the migrating MCCs and the distal appendage marker, while present, is not always localized fully around the circumference of this marker (Fig. 3 D), we cannot exclude the possibility that Dyrk1a and CEP97 are also involved in part in centriole maturation. This may not be unexpected, because Plk1 is known to be involved in centriole maturation (Al Jord et al., 2017).

An imbalance in the amount of Dyrk1a protein is implicated in a wide spectrum of diseases and deficits. Decreased Dyrk1a expression is recognized as a cause of some autism spectrum disorders, as well as microcephaly and intellectual disability in different species (Earl et al., 2017; van Bon et al., 2016). Recently, it was reported that loss of Dyrk1a also causes ciliogenesis defects in *Xenopus* MCCs (Willsey et al., 2020). It is unknown whether Dyrk1a mutations have a direct impact on ciliogenesis in the respiratory tract of patients with Down syndrome; however, cilia formation is disrupted in trisomy 21 cells derived from patients with Down syndrome (Galati et al., 2018). The effect on primary cilia has been reported to be likely due to overexpression of pericentrin and thus may affect protein trafficking to the centrosome via IFT20 (Galati et al., 2018). The *Dyrk1a* gene maps in the Down syndrome critical region, and recurrent respiratory infections are observed in patients (Santoro et al., 2021), suggesting the possibility that deregulated Dyrk1a expression may have an impact on multiciliogenesis and/or cilia function in the respiratory tract. In fact, respiratory tract infections are common in many patients with ciliopathy (Milla, 2016). Our findings on the collaborative role of CEP97 and Dyrk1a in multiciliation may add to our existing knowledge regarding pericentrin overexpression in patients with trisomy 21 and therapeutic approaches for the respiratory diseases in patients with ciliopathy and Down syndrome.

Taken together, our work suggests a novel mechanism by which CEP97 contributes to multiciliation in vertebrate MCCs through coordination with Dyrk1a and Plk1 (Fig. 8 C). The phosphorylation of CEP97 by Dyrk1a modulates the affinity of CEP97 for Plk1, which leads to mature centriole disengagement. Moreover, the association between CEP97 and Dyrk1a is critical for basal body migration and docking during ciliogenesis in MCCs, suggesting that CEP97 may play several roles in ciliogenesis, depending upon the stage of ciliogenesis, the cell context, and the cilia type (primary cilia, monocilia, and multicilia).

## Materials and methods

### *X. laevis* embryo microinjection and manipulation
*Xenopus* eggs were collected and in vitro fertilized using standard procedures (Sive et al., 2007). Embryos were microinjected with

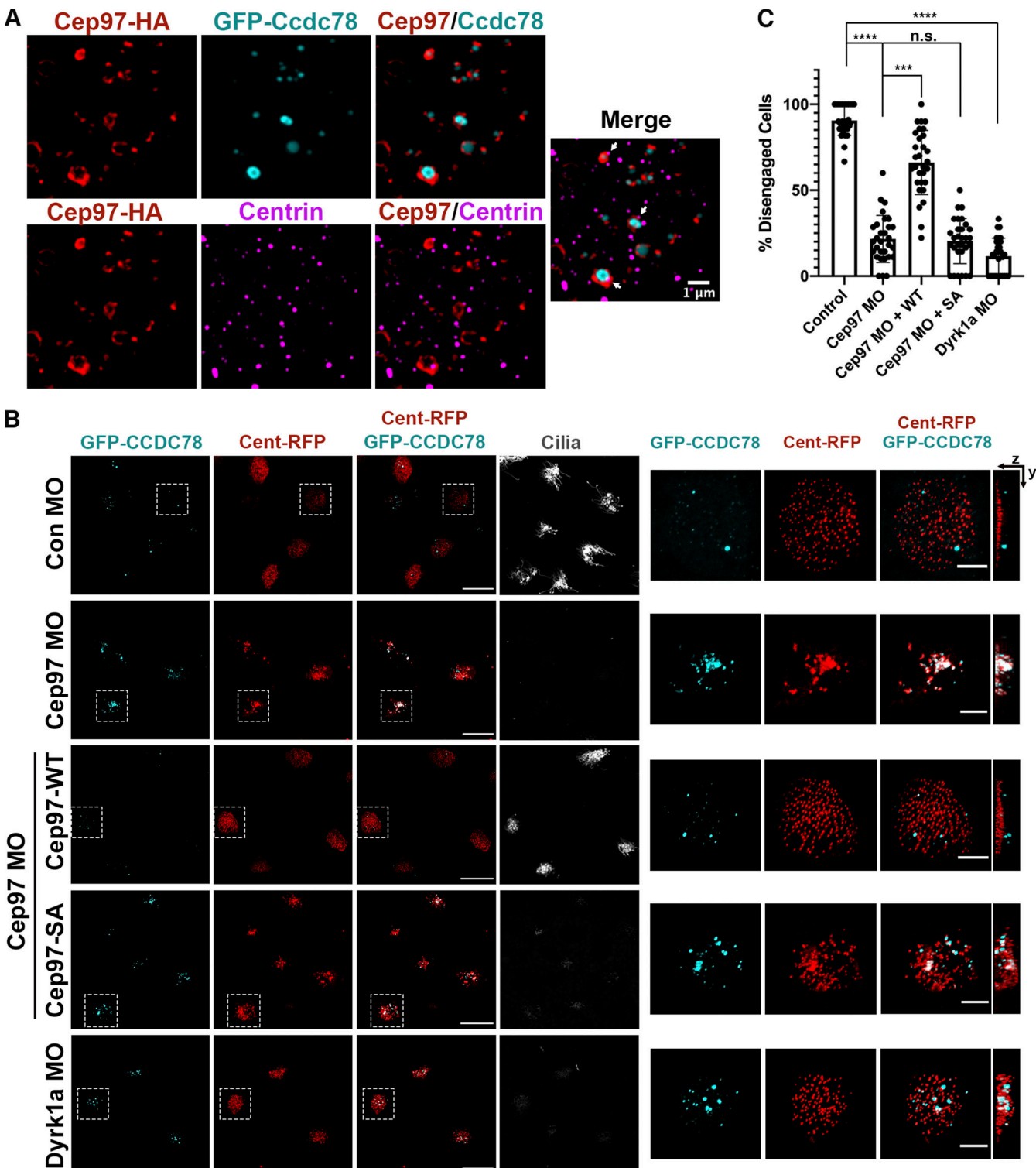

Figure 5. **CEP97-Dyrk1a modulates centriole disengagement. (A)** The subcellular localization of CEP97 to deuterosomes. The mRNAs of CEP97-HA and GFP-Ccdc78 (deuterosome marker) were injected into two ventral blastomeres at the eight-cell stage. Embryos were fixed at stage 19 and transverse sectioned, followed by immunofluorescence with HA probe (red), anti-GFP (cyan), and anti-centrin (magenta) antibodies. Images were generated by 3D-SIM. Scale bar, 1 µm. **(B)** Defective centriole disengagement upon knockdown of CEP97 and Dyrk1a. The indicated combinations of the mRNAs and MOs were injected into one blastomere of eight-cell stage embryos and fixed at stage 23. Cent-RFP marks basal bodies (red). GFP-Ccdc78 marks deuterosomes (cyan). Cilia are visualized by acetylated tubulin antibody staining. Images in white dotted square are magnified to the right. Rightmost images are 3D projections. Scale bars, large images = 20 µm; small images = 5 µm. Con, control. **(C)** Quantification of completed centriole disengagement in MCCs in B. Cent-RFP expressing MCCs in a field were considered for quantification. Graph is mean percentage ± SD of MCCs per image. Image numbers for quantification, *n* = 30; embryos per group from three independent experiments for each condition, *n* = 25; ***, P < 0.001; ****, P < 0.0001; one-way ANOVA.

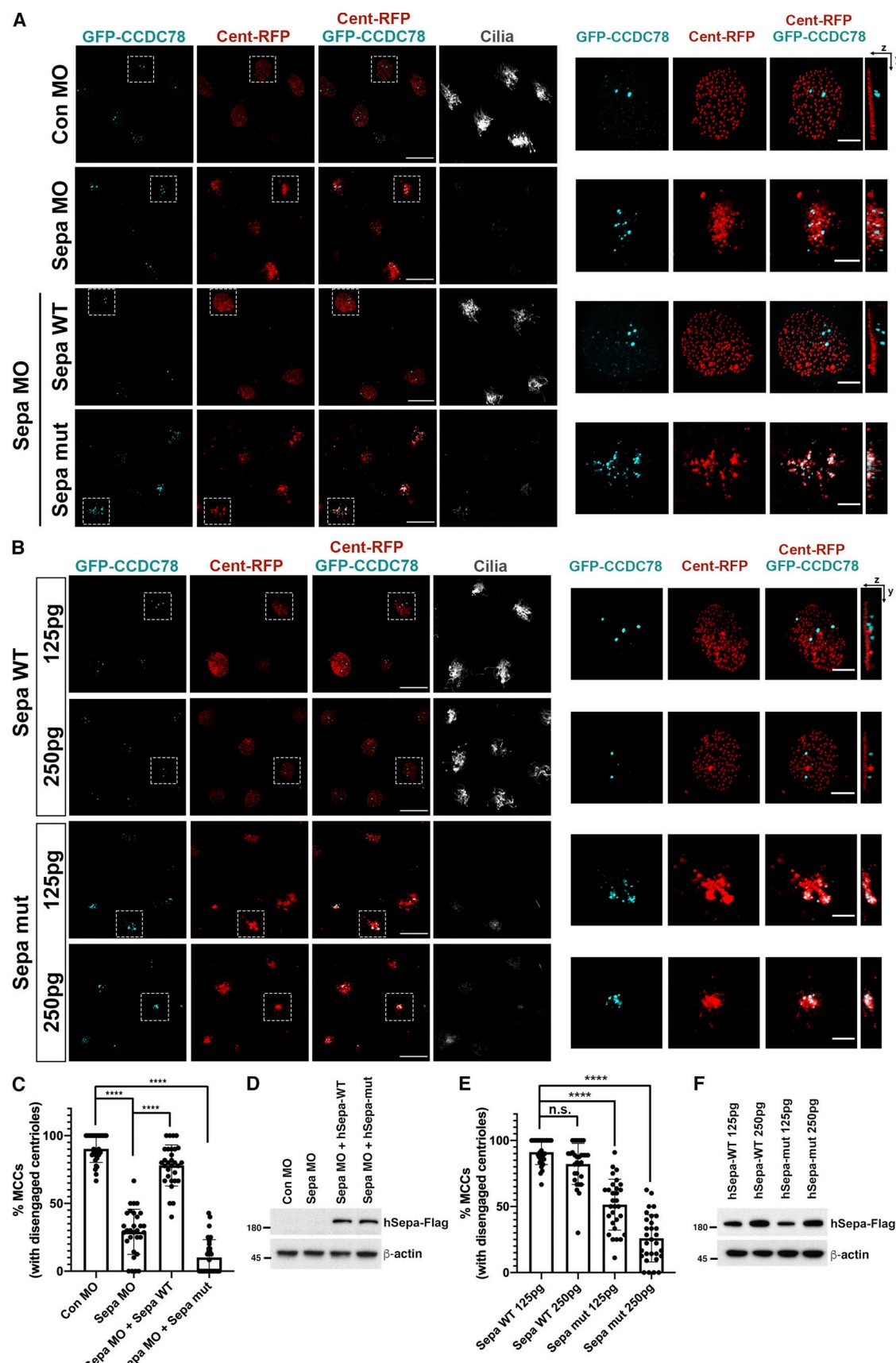

Figure 6. **Separase is required for centriole disengagement in *Xenopus* MCCs. (A and B)** Knockdown of Separase (Sepa; A) and expression of mutant Separase (B) caused centriole disengagement defects. The indicated mRNAs and MOs were injected into one ventral blastomere of eight-cell stage embryos,

and embryos were fixed at stage 23. GFP-Ccdc78, cyan; Cent-RFP, red. Antiacetylated tubulin antibody stains cilia. Images in white dotted square are magnified to the right. Rightmost images are 3D projections. Scale bars, large images = 20 µm; small images = 5 µm. **(C)** Quantification of completed centriole disengagement in MCCs in A. Graph is mean percentages ± SD of MCCs per image. Images for analysis, n = 30; embryos per group from three independent experiments, n = 20; ****, P < 0.0001; one-way ANOVA. **(D)** Immunoblot of the indicated exogenous proteins in A. Con, control. **(E)** Quantification of completed centriole disengagement in MCCs in B. Graph is mean percentage ± SD of MCCs per image. Images for analysis, n = 30; embryos per group from three independent experiments, n = 22; ****, P < 0.0001; one-way ANOVA. **(F)** Immunoblot of the indicated exogenous proteins in B. Numbers marked on left side of Western blot images represent protein molecular weight (kilodaltons).

the indicated mRNAs (0.5–1 ng) at one-cell stage for co-IPs. To target MCCs on the epidermis, one or both ventral blastomeres of four- or eight-cell stage embryos was injected with mRNAs (Cent-RFP 50 pg, Mem-GFP 200 pg, GFP-Ccdc78 125 pg, Deup1-GFP 30 pg, MO-resistant mRNAs 125–250 pg) and morpholinos (5–15 ng) for immunofluorescence. To target the GRP cilia, both dorsal blastomeres of eight-cell stage embryos were injected with mRNAs and morpholinos. Synthetic mRNAs were prepared using the SP6 mMESSAGE mMACHINE kit (AM1340; Life Technologies).

The MOs were purchased from Gene Tools: CEP97 MO: 5′-CCAAATGTGCCACTGCCATAGTAGC-3′; Dyrk1a MO: 5′-ATCGTCCTCTTTCAAGTCTCATATC-3′; Separase MO: 5′-AAGGACTTTATATTTGCTTCCATTC-3′; control MO: 5′-CCTCTTACCTCAGTTACAATTTATA-3′.

## Immunoblotting
TNSG buffer (50 mM Tris-HCl, pH 7.4, 150 mM NaCl, 1% NP-40, 10% glycerol, and protease inhibitors) was used for lysing embryos and cell lines. Lysates were loaded onto 8–10% SDS-polyacrylamide gels and transferred to polyvinylidene difluoride membrane, followed by blocking with 5% nonfat dry milk TBST buffer (TBS buffer with 0.1% Tween 20). Antibodies were incubated for 1 h at RT or overnight at 4°C. The following antibodies were used for immunoblotting: anti-CEP97 antibody (dilution 1:1,000, A301-947A; Bethyl Laboratories), anti-Dyrk1a antibody (dilution 1:1,000, ab69811; Abcam), anti–β-actin antibody (dilution 1:2,000, sc47778; Santa Cruz Biotechnology), anti-GAPDH antibody (1:2,000, sc-25778; Santa Cruz Biotechnology), anti–V5-peroxidase antibody (dilution 1:10,000, V2260; Sigma-Aldrich), anti-Flag-M2 HRP mAb (dilution 1:5,000, A8592; Sigma-Aldrich), and anti–HA-peroxidase mAb (dilution 1:5,000, H6533; Sigma-Aldrich).

## Molecular cloning and plasmids
*X. laevis* CEP97 (cDNA clone Mammalian Gene Collection [MGC]: 83642 IMAGE:5085089), *X. laevis* Dyrk1a (cDNA clone MGC: 132284 IMAGE:5542675), *X. laevis* Plk1 (cDNA clone MGC:52566 IMAGE:4930653), *Homo sapiens* CEP97 (cDNA clone MGC:45383 IMAGE:5528397), *H. sapiens* CDC20B (cDNA clone IMAGE: 5206729), and *H. sapiens* Spag5 (cDNA clone MGC:8476 IMAGE: 2821784) were PCR amplified and cloned into pCS2. HA, Flag, or V5 tag coding sequences were added to reverse primers to generate C-terminal tagged proteins. The indicated deletion mutants of CEP97 and Dyrk1a were generated by a site-directed mutagenesis method. For GFP- and mCherry-tagged proteins, inserts were subcloned to pCS2-GFP and pCS2-mCherry, respectively. WT human Separase (pcDNA5 FRT TO myc hSeparase, 59820) and inactive mutant (pcDNA5 FRT TO myc Separase

C2029A, plasmid 59822) were obtained from Addgene and subcloned to pCS2 by PCR amplification with a Flag tag coding sequence at the C-terminus. Full-length *Xenopus* Dvl2 was previously described (Lee et al., 2019). Full-length xCcdc78 (Xl. 4890) was PCR amplified from stage 17 *Xenopus* cDNA as previously described (Klos Dehring et al., 2013), followed by subcloning to produce N-terminal GFP-Ccdc78. CLAMP-GFP and Cent-RFP plasmids were a generous gift from Dr. John Wallingford (The University of Texas at Austin, Austin, TX). The Deup1-GFP plasmid was a generous gift from Dr. Brian Mitchell (Northwestern University, Chicago, IL). All plasmids used for this study were verified by DNA sequencing. See the primer sequences used for this study in Table S3.

## Co-IP
The injected embryos were harvested at stages 11–12 and lysed with TNSG buffer containing protease inhibitors (cOmplete Protease inhibitor cocktail [Sigma-Aldrich] and PMSF) and phosphatase inhibitors (sodium orthovanadate and β-glycerophosphate). Exogenous tagged proteins were pulled down with anti-V5 agarose beads (A7345; Sigma-Aldrich), anti-Flag M2 affinity gel (A2220; Sigma-Aldrich), or HA epitope tag antibody agarose conjugate (sc-500777A; Santa Cruz Biotechnology) for 6–8 h at 4°C on a nutator, followed by washing with TNSG lysis buffer. For endogenous protein co-IPs, HEK293T cells were lysed with NP-40 lysis buffer (25 mM Hepes, 150 mM KCl, 1.5 mM MgCl$_2$, 0.5% NP-40, 5% glycerol, and protease and phosphatase inhibitors). 1 mg of total lysates were incubated with 1 µg of Dyrk1a antibody (ab69811; Abcam) at 4°C overnight, followed by an additional incubation with protein A/G plus agarose beads (sc-2003; Santa Cruz Biotechnology) for 2 h and washing with the lysis buffer four times. Co-IPs were repeated at least three times.

## Immunofluorescence and microscopy
Immunofluorescence was performed as previously described with minor modifications (Lee et al., 2008; Werner and Mitchell, 2013). Briefly, embryos were fixed with MEMFA (1× MEM salt containing 4% formaldehyde at RT for 2 h or at 4°C overnight) or Dent's fixative (80% methanol and 20% DMSO at 4°C overnight). Fixed embryos were washed with PBST (1× PBS, pH 7.4, 0.1% Triton X-100) three times, followed by incubation with blocking buffer (PBST containing 2% BSA). The embryos were nutated at 4°C overnight in blocking buffer containing the indicated antibody. The following antibodies were used for immunofluorescence: acetylated tubulin–Alexa Fluor 647–conjugated antibody (dilution 1:1,000, anti-acetylated tubulin, T7451 [Sigma-Aldrich]; Zip Alexa Fluor 647 Rapid Antibody Labeling Kit, Z11235 [Thermo Fisher Scientific]), antiacetylated tubulin mAb

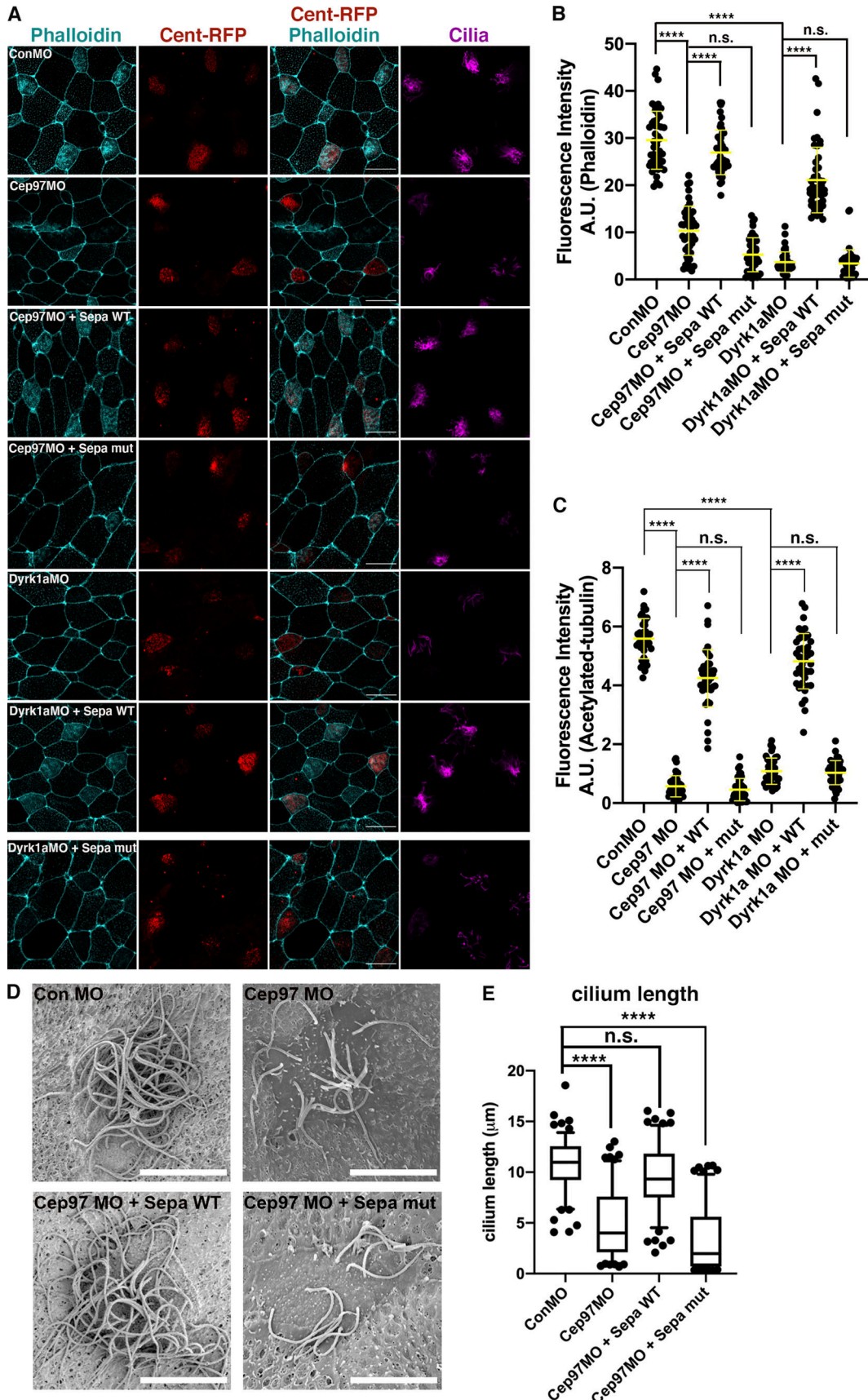

Figure 7. **Human Separase expression rescues impaired ciliogenesis in CEP97 and Dyrk1a morphant MCCs. (A)** The embryos injected with the indicated mRNAs and MOs were fixed at stage 26. Cent-RFP (red) marks basal bodies. Multicilia are visualized by antiacetylated tubulin antibody. Phalloidin (cyan) stains

apical F-actin. Scale bars, 20 µm. **(B)** Quantification of phalloidin intensity in MCCs in A. ****, P < 0.0001; one-way ANOVA; MCCs, n = 46; embryos per group from one representative of three independent experiments, n = 8. Data are mean ± SD. Con, control; Sepa, Separase. **(C)** Quantification of relative acetylated tubulin intensity in A. Number of field images for analysis, n = 40 from 20 embryos for each condition; ****, P < 0.0001; one-way ANOVA. Error bars indicate ±SD. **(D)** SEM of embryonic epidermis at stage 28. The indicated mRNAs and Mos were injected into both ventral blastomeres at the eight-cell stage. Scale bars, 7.5 µm. **(E)** Quantification of cilia length in MCCs. Measured cilia, n > 100; embryos per group, n = 3; ****, P < 0.0001; one-way ANOVA. Error bars indicate ±SD.

(dilution 1:1,000, T7451; Sigma-Aldrich), anti-GFP antibody (dilution 1:500, ab13970; Abcam), chicken anti-GFP antibody (dilution 1:500, NB100-1614; Novus Biologicals), anti-Cep164 antibody (dilution 1:500, 22227-1-AP; ProteinTech), and anti-centrin antibody (dilution 1:100, 04-1624; EMD Millipore). For apical actin staining, the fixed embryos were incubated with Alexa Fluor 488 phalloidin (dilution 1:250, A12379; Thermo Fisher Scientific). Confocal microscopy was conducted with a Zeiss LSM 880 laser scanning confocal microscope equipped with a Plan Apochromat 63×/1.4 NA oil differential interference contrast objective. Serial z-stack images were taken with 0.2-mm distance for most experiments but with 0.1-mm distance for imaging centriole disengagement (Figs. 5, 6, and 7). Immunostained embryos were imaged using identical settings for all samples in an individual experiment. For example, the images shown in Fig. 5 B were generated with this setting: 488 channel (gain [master] 850, digital offset 33, digital gain 1.0), 568 channel (gain [master] 850, digital 45, digital gain 1.3), and 647 channel (gain [master] 700, digital 150, digital gain 1.0). All images were processed under identical conditions.

For colocalization studies in Figs. 4 A and 5 A and Fig. S3, C and D, a transverse section was performed at 60-µm thickness using a Leica VT 1200S vibratome (Leica Microsystems).

SIM was operated on an N-SIM microscope (Nikon Instruments) equipped with an Apo total internal reflection fluorescence 100×/1.49 NA Plan Apo oil objective and a back-illuminated 16-µm pixel EM charge-coupled device camera (DU897; Andor). Z-stacks with 0.1-µm Z-distance were acquired in 3D-SIM mode, and super-resolution images were generated by reconstruction of raw images. The XYZ shift was corrected using 0.2-µm TetraSpeck microspheres (T7280; Thermo Fisher Scientific). All whole-mount antibody-stained embryos and sectioned tissues were imaged by identical imaging settings for all groups of samples in one experiment, and all images were processed under identical conditions.

**Recombinant protein preparation and in vitro kinase assay**
Recombinant CEP97 and Dvl2 proteins were prepared using the TnT SP6 High-Yield Wheat Germ Protein Expression System (L3260; Promega) according to the manufacturer's instructions for the CEP97 protein shift assay. For in vitro kinase assay–MS, recombinant CEP97 protein extracted and purified from bacteria was used. Briefly, the pGEX4T-2 plasmids encoding aa 1–420 and 421–807 of CEP97 were transformed into Rosetta (DE3)-competent cells (70954; Sigma-Aldrich). Bacterial lysates were incubated with glutathione Sepharose 4B resin for 3 h. After washing resin with washing buffer 1 (PBS including 1% Triton X-100) twice, washing buffer 2 (50 mM Tris, pH 8.5, 150 mM NaCl) twice, and elution buffer (50 mM Tris, pH 8.5, 150 mM NaCl,

2.5 mM $CaCl_2$) twice, the resin was incubated with thrombin protease (GE27-0846-01; Sigma-Aldrich) for 18 h at RT. The cleaved CEP97 proteins were used for in vitro kinase assay–MS. The in vitro kinase assay was performed according to the manufacturer's instructions. Briefly, the mixture of 10 µg of purified recombinant CEP97, 1 µl of active Dyrk1a (K23-09; SignalChem Biotech), and 200 µM ATP in kinase assay buffer (K01-09; SignalChem Biotech) were incubated for 1 h at 30°C, and MS (Poochon Scientific) was performed.

**EM**
Embryos were fixed in 4% formaldehyde + 2% glutaraldehyde in 0.1 M sodium cacodylate buffer overnight at 4°C. On the following day, embryos were washed three times with 0.1 M sodium cacodylate buffer and stained with 1% osmium tetroxide and 0.5% uranyl acetate for 1 h each. Then, a series of ethanol solutions (35%, 50%, 70%, 95%, and 100%) were applied for dehydration. For SEM, samples were further dehydrated using tetramethylsilane three times for 10 min each and air dried. For TEM, samples were immersed in Embed 812 resin overnight after ethanol dehydration to allow complete infiltration and placed in a 55°C oven for 48 h to finish resin curing. Resin blocks were thin sectioned (70–80 nm) and mounted on 150-grid mesh, and the sections were counterstained in aqueous uranyl acetate (0.5%) and Reynolds lead citrate. Images were captured on a Hitachi S4500 SEM and a Hitachi H7600 TEM, respectively.

**Nanospray liquid chromatography with tandem MS (LC/MS-MS) analysis and database search**
The immunoprecipitated proteins from IP bead pellets were eluted by 4% SDS followed by in-solution trypsin digestion. The digests were purified by C18-Zip-tip before analysis by using LC/MS-MS. LC/MS-MS analysis was performed on a Q Exactive Hybrid Quadrupole-Orbitrap Mass Spectrometer and a Dionex UltiMate 3000 RSLCnano System (Thermo Fisher Scientific). Each peptide fraction was loaded onto a peptide trap cartridge at a flow rate of 5 µl/min. The trapped peptides were eluted onto a reversed-phase 20-cm C18 PicoFrit column (New Objective) using a linear gradient of acetonitrile (3–36%) in 0.1% formic acid. The elution duration was 60 min at a flow rate of 0.3 µl/min. Eluted peptides from the PicoFrit column were ionized and sprayed into the mass spectrometer using a Nanospray Flex Ion Source (ES071; Thermo Fisher Scientific) under the following settings: spray voltage 1.8 kV and capillary temperature 250°C. The Q Exactive Hybrid Quadrupole-Orbitrap Mass Spectrometer was operated in the data-dependent mode to automatically switch between full-scan MS and MS/MS acquisition. Survey full-scan MS spectra (mass-to-charge ratio, 300–2,000) was acquired in the Orbitrap with 35,000 resolutions (mass-to-charge

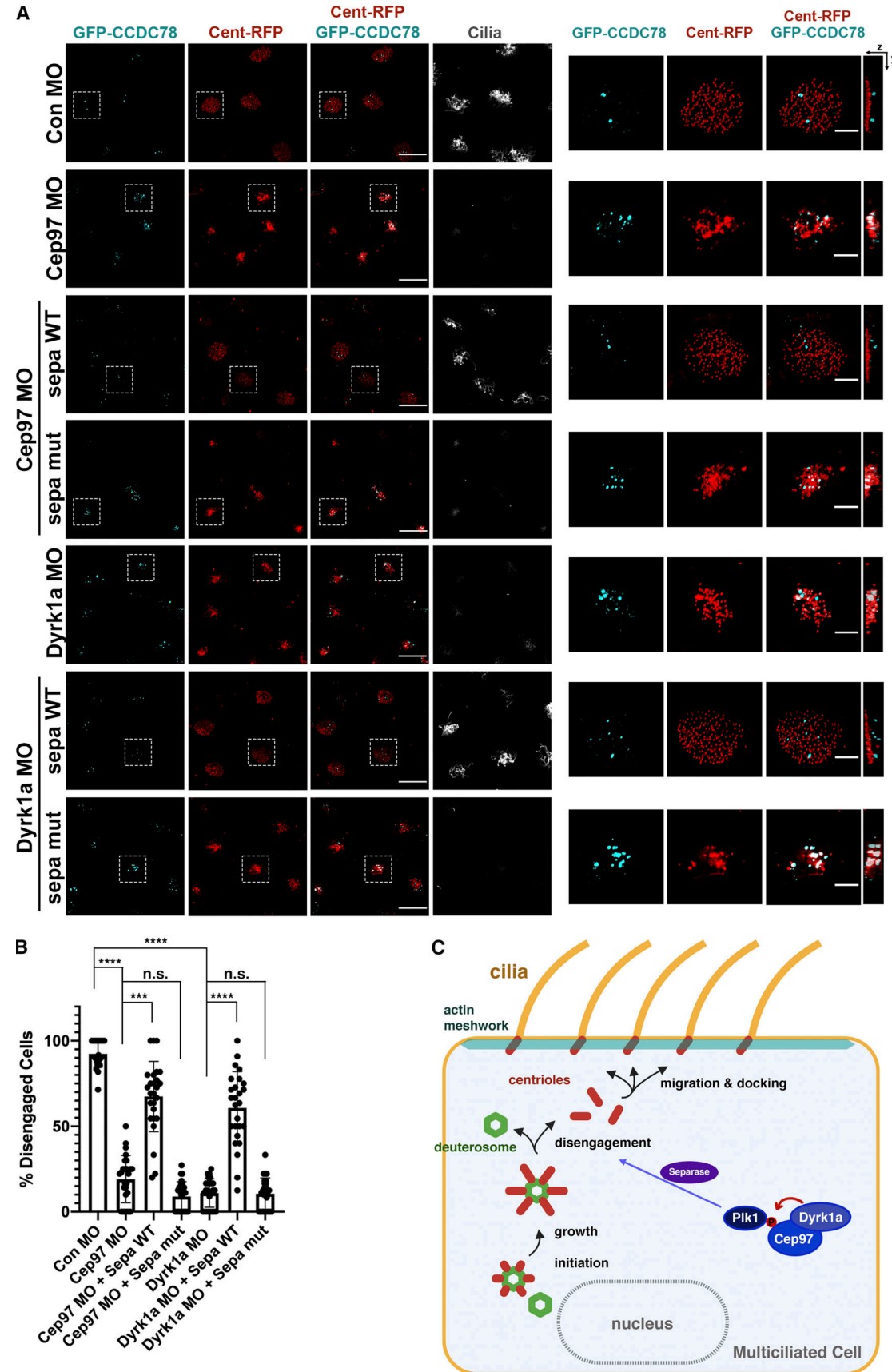

**Figure 8. Human Separase expression restores defective centriole disengagement in CEP97 and Dyrk1a morphant MCCs. (A)** The embryos injected with the indicated mRNAs and MOs were fixed at stage 23, and then immunofluorescence was performed. Cent-RFP (red) marks basal bodies. GFP-Ccdc78

(cyan) marks deuterosomes. Multicilia are visualized by antiacetylated tubulin antibody. Images in dotted square are magnified to the right. Rightmost images are 3D projections. Scale bars, large images = 20 µm; small images = 5 µm. Con, control; sepa, Separase. **(B)** Quantification of the completed centriole disengagement in A. Cent-RFP expressing MCCs in a field were considered for quantification. Bar graphs display the mean percentage of disengaged MCCs ± SD. Number of field images for measurement per group, n = 26; embryos per group, n = 18; ****, P < 0.0001; ***, P < 0.0001; one-way ANOVA. **(C)** Working model. The phosphorylation of CEP97 by Dyrk1a modulates the affinity of CEP97 for Plk1, which leads to mature centriole disengagement, basal body migration, and docking during ciliogenesis in MCCs. Centrioles in red, cilia in orange, actin meshwork in light green, and deuterosome in green.

ratio, 200) after an accumulation of ions to a 3 × 10e6 target value based on predictive automatic gain control. The maximum injection time was set to 100 ms. The 15 most intense multiply charged ions (z ≥ 2) were sequentially isolated and fragmented in the octopole collision cell by higher-energy collisional dissociation using normalized higher-energy collisional dissociation collision energy 30 with an automatic gain control target of 1 × 10e5 and a maximum injection time of 100 ms at 17,500 resolutions. The isolation window was set to 2. The dynamic exclusion was set to 20 s. Charge state screening was enabled to reject unassigned and 1+, 7+, 8+, and >8+ ions.

### MS peptide assignment

MS raw data files were searched against the UniProt Knowledgebase (UniProtKB) *X. laevis* protein sequence database including the CEP97 sequence using the Proteome Discoverer Software version 1.4 (Thermo Fisher Scientific) based on the SEQUEST and percolator algorithms. The searches were performed using a false discovery rate of 0.01. Carbamidomethylation (+57.021 D) of cysteines was a fixed modification, and Oxidation Met and Deamidation Q/N-deamidated (+0.98402 D), Methyl/+14.016 D (K, R), Acetyl/+42.011 D (K), Phospho/+79.966 D (S, T, Y), and Dimethyl/+28.031 D (K, R) were set as dynamic modifications. The resulting Proteome Discoverer report contains all assembled proteins with peptide sequences and possible phosphorylation modifications on S/T/Y and peptide spectrum match counts and MS1 ion peak area.

### Quantification and statistical analysis

Statistical analyses were performed using GraphPad Prism version 8 software. The quantification method is described in the figure legends. Control and experimental groups were compared by one-way ANOVA with Tukey's multiple comparisons test or unpaired two-tailed *t* test. Some parametric tests were performed without the normality test because data distribution was assumed to be normal.

### Online supplemental material

Fig. S1 shows morpholino target sequence information and Crispr/Cas9-mediated CEP97 knockout phenotypes in *Xenopus* MCCs. Fig. S2 shows CEP97 and Dyrk1a knockdown effects on MCCs and GRPs. Fig. S3 shows CEP97 and Plk1 localization in MCCs and CEP97 interactions with Cdc20B and Spag5 in HEK293T cells. Fig. S4 shows CEP97 and Plk1 localization in the absence of Dyrk1a and CEP97 using 3D-SIM. Fig. S5 shows that CEP97 and Dyrk1a knockdown causes centriole disengagement defects at different stages. Table S1 lists potential binding proteins of CEP97 from IP-MS. Table S2 lists in vitro phosphorylation sites

of CEP97 by active Dyrk1a. Table S3 contains primer sequence information used for cloning.

## Acknowledgments

We thank K. Peifley and V. Magidson for technical support with microscopy and all members of the Cancer and Developmental Biology Laboratory for discussion and comments.

This research was supported in part by the Intramural Research Program of the National Institutes of Health, National Cancer Institute. The content of this publication does not necessarily reflect the views or policies of the U.S. Department of Health and Human Services, nor does mention of trade names, commercial products, or organizations imply endorsement by the U.S. government.

The authors declare no competing financial interests.

Author contributions: M. Lee designed and carried out the experiments with the help of K. Nagashima, Y.-S. Hwang, J. Yoon, J. Sun, Z. Wang, C. Carpenter, and H.-K. Lee. K. Nagashima, C. Carpenter, and Z. Wang assisted with EM. C.J. Westlake contributed to data analysis and interpretation and edited the manuscript. M. Lee and I.O. Daar wrote the manuscript. I.O. Daar supervised the project. All of the authors discussed the results and commented on the manuscript.

Submitted: 18 February 2021

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

# Supplemental material

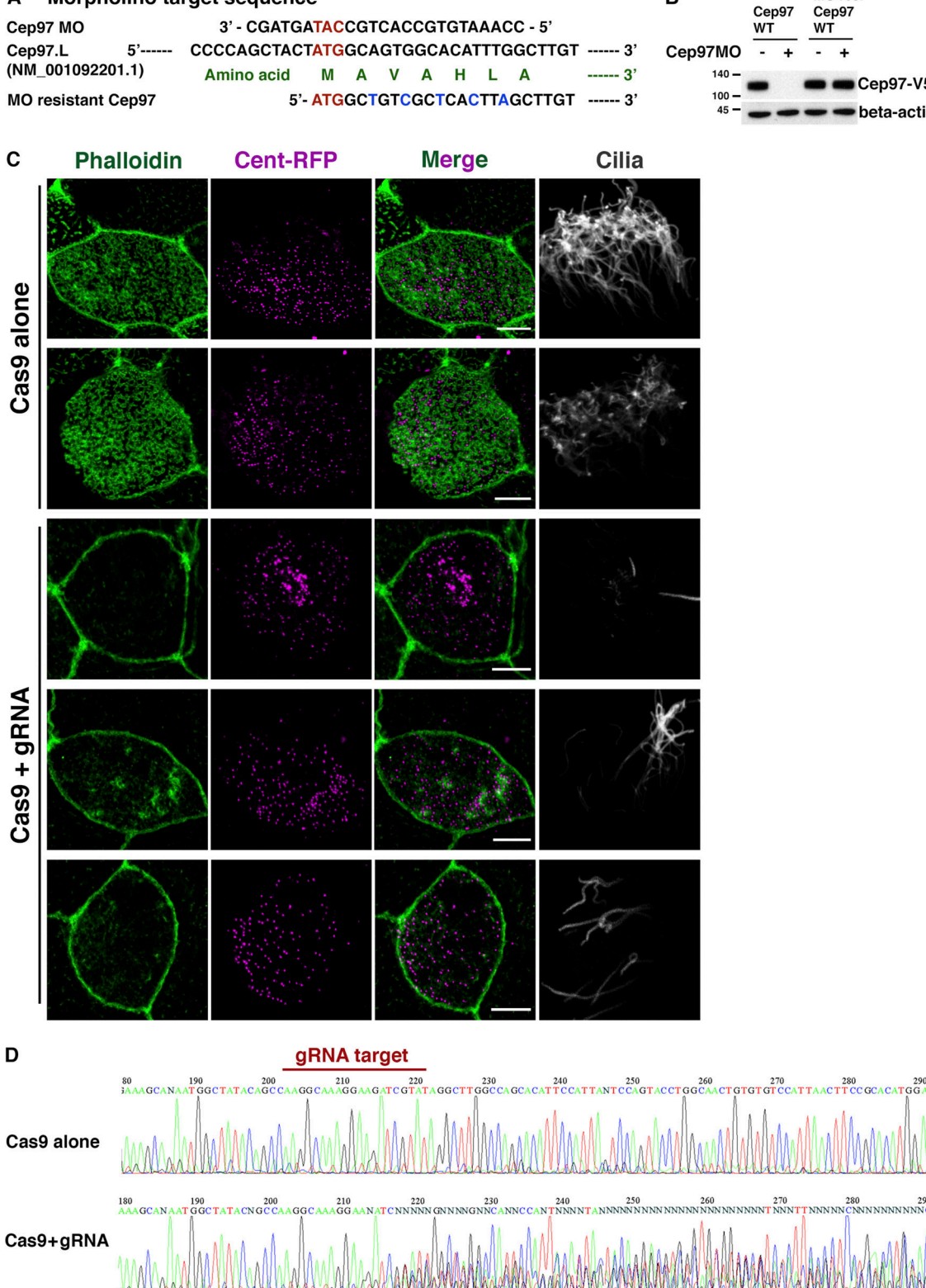

Figure S1. **CEP97 knockout causes ciliogenesis defects in MCCs. (A)** The antisense MOs and morpholino-resistant CEP97 construct nucleotide sequence are aligned with the *X. laevis* CEP97 target sequence. **(B)** The verification of morpholino knockdown efficacy. Immunoblot shows morpholino-resistant (MO resi) CEP97 expression in morpholino coinjected embryos, but WT CEP97 expression is compromised by morpholinos. Numbers marked on left side of Western blot images represent protein molecular weight (kilodaltons). **(C)** CEP97 knockout phenocopies CEP97 knockdown effect on MCC cilia formation. Cas9 protein with or without CEP97 targeting gRNA was coinjected with Cent-RFP mRNA to one ventral blastomere of four-cell stage embryos, and the embryos were fixed at stage 27. Cilia was visualized by immunofluorescence using acetylated tubulin antibody. Phalloidin (green) marks apical F-actin. Scale bars, 5 µm. **(D)** Confirmation of CEP97 genetic disruption by DNA-sequencing data using genomic DNAs of embryos in C.

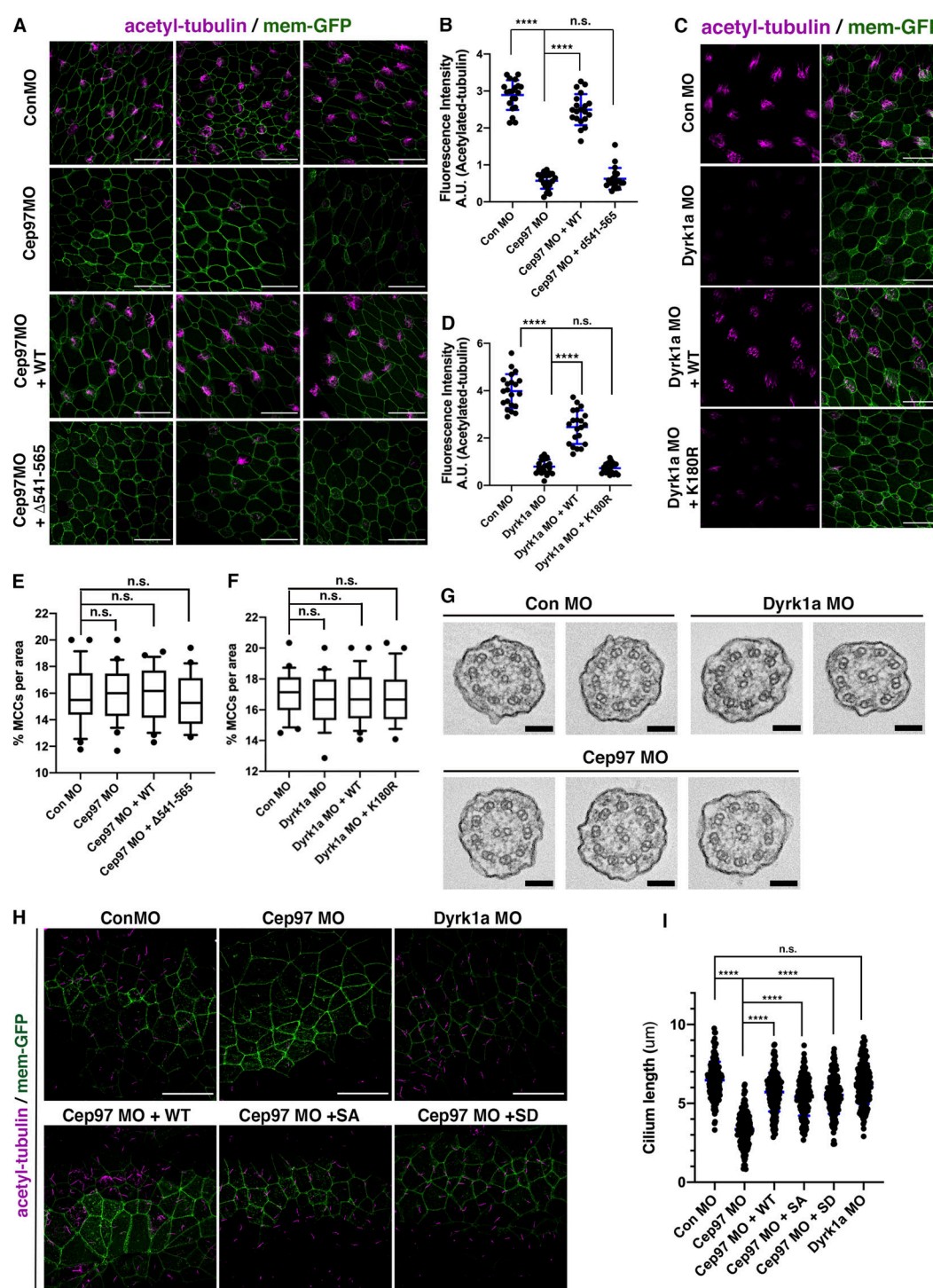

Figure S2. **CEP97 and Dyrk1a are required for ciliation in MCCs, and CEP97 is required in GRP. (A and C)** A cocktail of the indicated mRNAs and MOs was injected into one ventral blastomere of four-cell stage embryos and fixed at stage 27. Acetylated tubulin staining (magenta) marks cilia. Mem-GFP (green) was used as a tracer. Scale bars, 50 μm. Con, control. **(B)** Quantification of relative acetylated tubulin intensity in A (image numbers for analysis, n = 30; embryos per group from three independent experiments for each condition, n = 16). ****, P < 0.0001; one-way ANOVA. Error bars indicate ±SD. **(D)** Quantification of relative acetylated tubulin intensity in C (image number for analysis, n = 25; embryos per group from three independent experiments, n = 14). ****, P < 0.0001; one-way ANOVA. Error bars indicate ±SD. **(E and F)** Quantification of the total percentage of MCCs in A and C. No significant difference is observed in the percentage of total MCCs in CEP97 and Dyrk1a morphants. Image numbers for analysis, n = 25; one-way ANOVA. **(G)** TEM of multicilia in control, CEP97, and Dyrk1a morphant MCCs. Cross-sectioned TEM images do not show any structural defects in axonemes on CEP97 and Dyrk1a knockdown. Scale bars, 100 nm. **(H)** CEP97 knockdown, but not Dyrk1a knockdown, decreases the length of GRP cilia. A cocktail of the indicated mRNAs and MOs was injected into two dorsal blastomeres of eight-cell stage embryos and fixed at stage 16. Acetylated tubulin staining (magenta) marks GRP cilia. Mem-GFP (green) was used as a tracer. Scale bars, 50 μm. **(I)** Quantification of GRP cilia length in H. Measured cilia number per group, n > 800; dissected GRPs per group from three independent experiments, n = 20. ****, P < 0.0001; one-way ANOVA. Error bars indicate ±SD. A.U., arbitrary units.

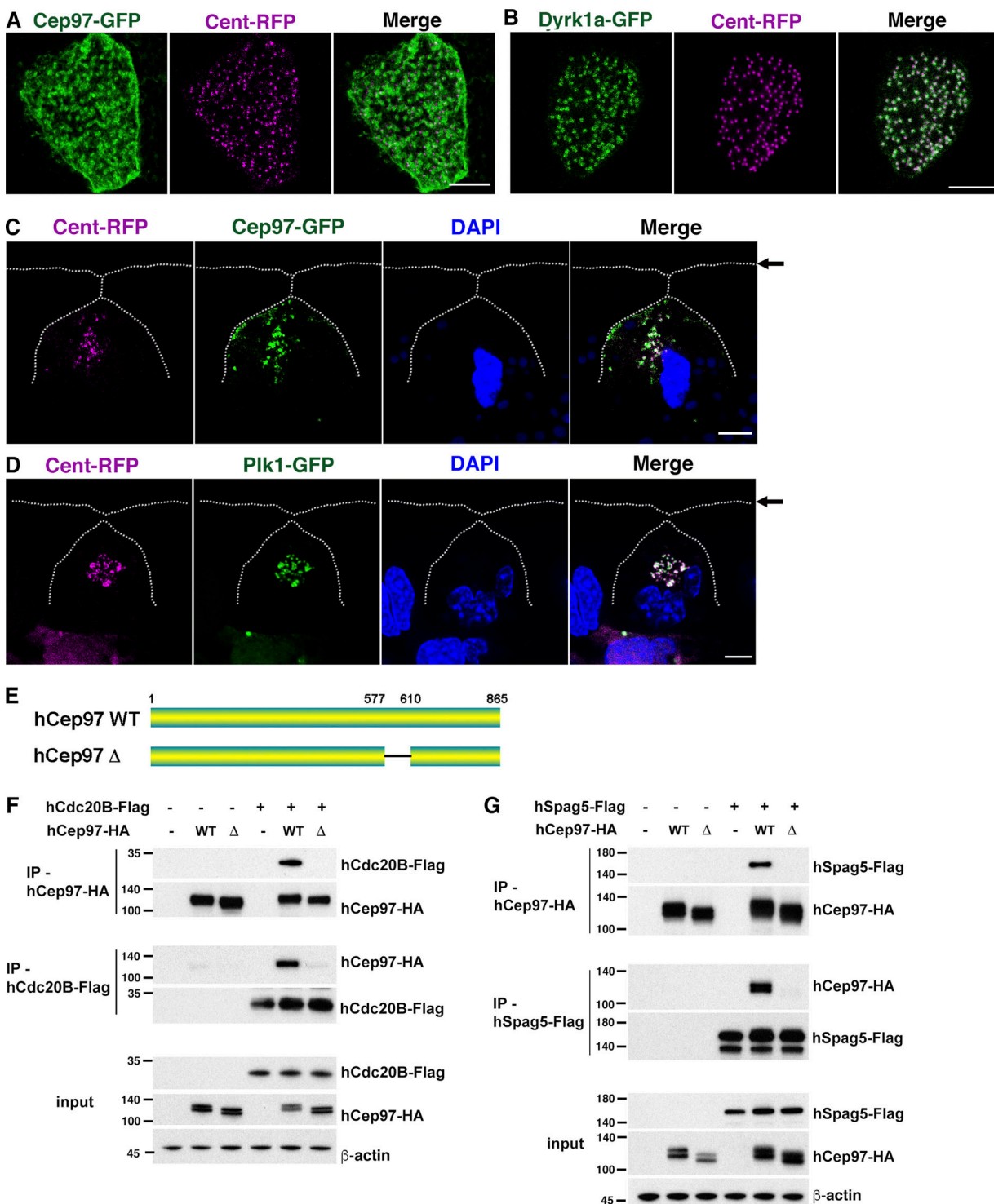

Figure S3. **CEP97 localizes to deuterosome and associates with Cdc20B and Spag5. (A and B)** CEP97 localizes near the basal body area (Cent-RFP) to an extent; yet CEP97 broadly localizes to the apical region of MCCs. Dyrk1a colocalizes with Cent-RFP. The indicated mRNAs were injected, and the embryos were fixed at stage 23 for confocal imaging. Scale bars, 5 µm. **(C and D)** The subcellular localization of CEP97 and Plk1 in migrating MCCs. The indicated synthetic mRNAs were injected into one ventral blastomere of four-cell stage embryos. The embryos were fixed at stage 19 and transverse sectioned. Cent-RFP, magenta; CEP97-GFP and Plk1-GFP, green. DAPI stains nuclei. Arrows mark the apical surface. Scale bars, 5 µm. **(E)** Schematic diagram of human CEP97 WT and deletion mutant (Δ577–610, corresponding to *Xenopus* CEP97 Δ541–565). **(F and G)** The indicated DNAs were transfected into HEK293T cells, and reciprocal co-IPs were performed. WT hCEP97-HA, but not Δ577-610 mutant-HA, associates with hCdc20B-Flag (F) and hSpag5-Flag (G). Numbers marked on left side of Western blot images represent protein molecular weight (kilodaltons).

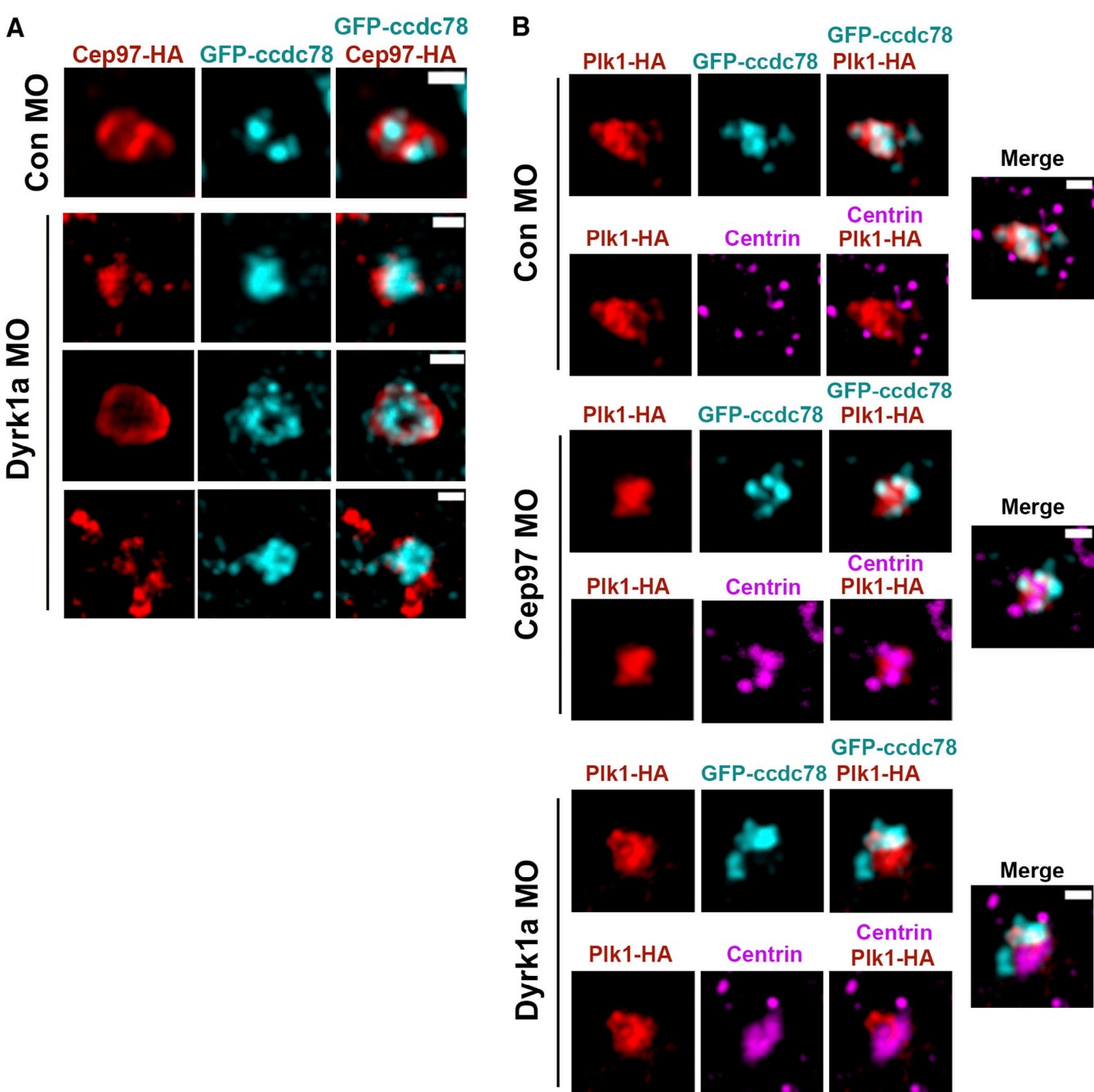

Figure S4. **CEP97 and Dyrk1a do not affect one another's deuterosome localization. (A)** Dyrk1a knockdown does not affect CEP97 localization to deuterosome. The embryos injected with the indicated mRNAs and MOs were fixed at stage 19 and sectioned, followed by immunostaining and 3D-SIM. Cep97-HA, red; GFP-CCDC78, cyan. Scale bars, 500 nm. Con, control. **(B)** Loss of CEP97 and Dyrk1a does not change Plk1 localization to deuterosome. Embryos were fixed at stage 19 and sectioned. Plk1-HA, red; GFP-CCDC78, cyan; centrin, magenta. Images were generated by 3D-SIM. Scale bars, 500 nm.

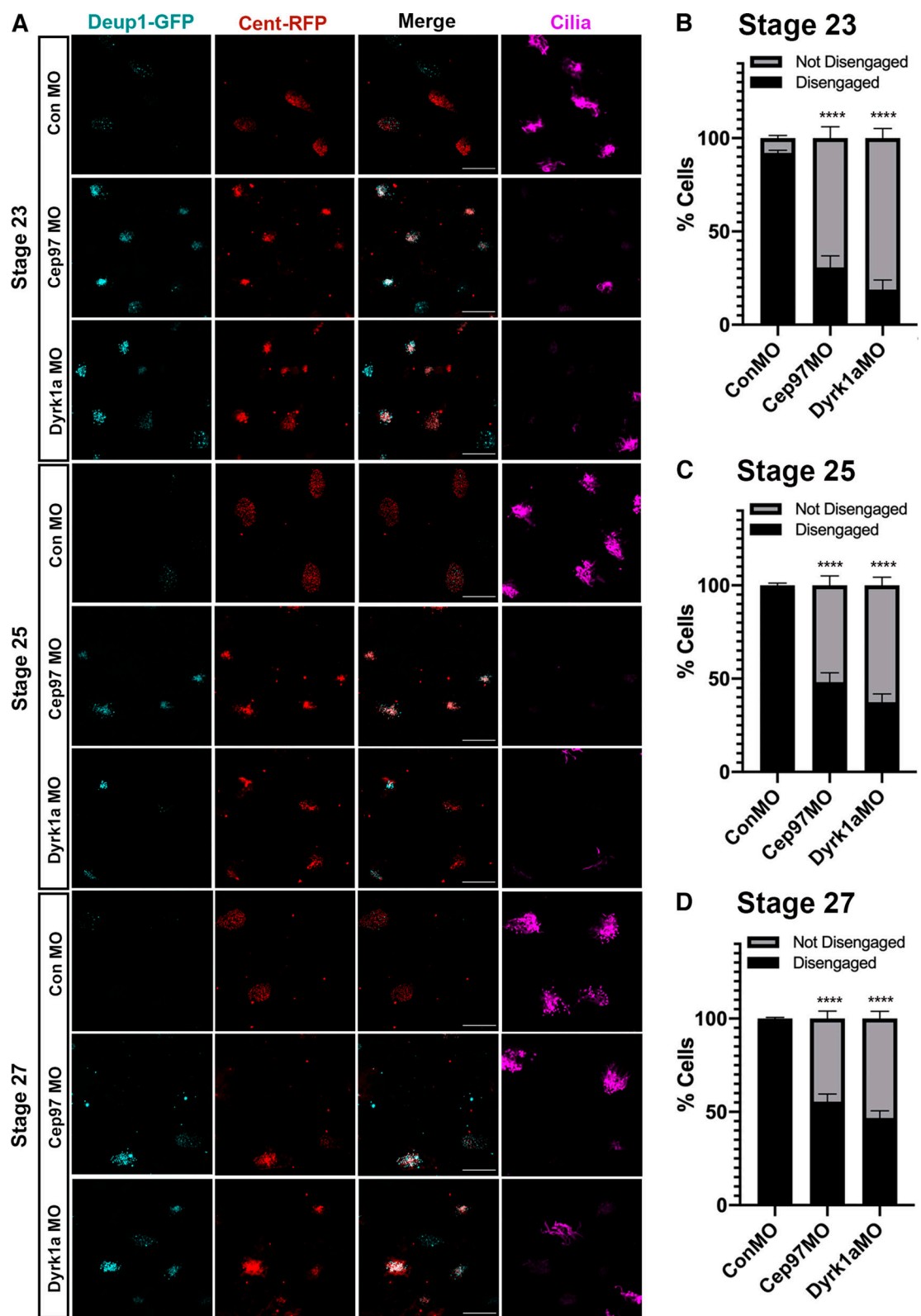

Figure S5. **CEP97 and Dyrk1a are required for centriole disengagement in *Xenopus* MCCs. (A)** Although defective centriole disengagement is relieved to some extent during embryonic development, a significant number of MCCs in CEP97 (46%) and Dyrk1a (55%) morphants still show incomplete centriole disengagement at stage 27. The cocktail of the mRNAs and MOs was injected into one ventral blastomere of eight-cell stage embryos and fixed at stages 23, 25, and 27. Cent-RFP (red) marks basal bodies. Deup1-GFP (cyan) marks deuterosomes. Cilia are visualized by acetylated tubulin antibody staining. Scale bars, 20 µm. Con, control. **(B–D)** Quantification of MCCs that completed or failed to complete centriole disengagement in A. Cent-RFP expressing MCCs in a field were considered for quantification. Bar graphs display the mean percentage of disengaged MCCs ± SE. Number of field images for measurement per group, *n* = 20; embryos per group from three independent experiments, *n* = 20; ****, P < 0.0001; one-way ANOVA.

**Provided online are three tables. Table S1 lists potential binding proteins of CEP97 from IP-MS. Table S2 lists in vitro phosphorylation sites of CEP97 by active Dyrk1a. Table S3 contains primer sequence information used for cloning.**

