## [Peer Review File · The Journal of Cell Biology]

CEP97 phosphorylation by Dyrk1a is critical for centriole separation during multiciliogenesis

moonsup lee, Kunio Nagashima, Jaeho Yoon, Jian Sun, Ziqiu Wang, Christina Carpenter, Hyun-Kyung Lee, Yoo-Seok Hwang, Christopher Westlake, and Ira Daar

Corresponding Author(s): Ira Daar, National Cancer Institute

Review Timeline:

Submission Date:	2021-02-18
Editorial Decision:	2021-03-22
Revision Received:	2021-08-18
Editorial Decision:	2021-09-20
Revision Received:	2021-09-27

Monitoring Editor: Monica Bettencourt-Dias

Scientific Editor: Lucia Morgado-Palacin

Transaction Report:

DOI: <https://doi.org/10.1083/jcb.202102110>

March 22, 2021

Re: JCB manuscript #202102110

Dr. Ira Daar
National Cancer Institute
Cancer & Developmental Biology Laboratory
Bld. 560 rm. 12-88
Frederick, MD 21702

Dear Dr. Daar,

Thank you for submitting your manuscript entitled "Dyrk1a phosphorylation of CEP97 is critical for centriole separation during multiciliogenesis". Your manuscript has been assessed by expert reviewers, whose comments are appended below. Although the reviewers express potential interest in this work, significant concerns, unfortunately, preclude publication of the current version of the manuscript in JCB.

You will see that the reviewers share important experimental and conceptual concerns. They highlight issues with data interpretation based on the reagents and tools, such as the Deu1-GFP line. Importantly, the reviewers ask for further analyses of the mechanism by which Cep97 controls centriole assembly in MCCs (e.g., Rev#2 points #1, #5, Rev#3 point #5). The referees ask for stronger binding studies to understand how Plk1 binds to Cep97 and stronger localization studies (Rev#2, Rev#3 #3, requesting stronger evidence that Cep97 is a scaffold between Dyrk1a and Plk1; Rev#2 #4 - Rev#3 #4). Both revs #2-3 ask for the MS data for Cep97 interactors to be included in the paper, which we agree is important. We overall agree with the referees that additional experimental efforts are needed to confirm the key phenotypes, bolster the localization and binding studies, as well as further strengthen the link between Cep97 and centriole assembly along the lines suggested by the reviewers.

Please let us know if you are able to address the major issues outlined above and wish to submit a revised manuscript to JCB. Note that a substantial amount of additional experimental data likely would be needed to satisfactorily address the concerns of the reviewers. As you may know, the typical timeframe for revisions is three to four months. However, we at JCB realize that the implementation of social distancing and shelter in place measures that limit spread of COVID-19 also pose challenges to scientific researchers. Lab closures especially are preventing scientists from conducting experiments to further their research. Therefore, JCB has waived the revision time limit. We recommend that you reach out to the editors once your lab has reopened to decide on an appropriate time frame for resubmission. Please note that papers are generally considered through only one revision cycle, so any revised manuscript will likely be either accepted or rejected.

If you choose to revise and resubmit your manuscript, please also attend to the following editorial points. Please direct any editorial questions to the journal office.

GENERAL GUIDELINES:

Text limits: Character count is < 40,000, not including spaces. Count includes title page, abstract, introduction, results, discussion, acknowledgments, and figure legends. Count does not include

materials and methods, references, tables, or supplemental legends.

Figures: Your manuscript may have up to 10 main text figures. To avoid delays in production, figures must be prepared according to the policies outlined in our Instructions to Authors, under Data Presentation, <https://jcb.rupress.org/site/misc/ifora.xhtml>. All figures in accepted manuscripts will be screened prior to publication.

IMPORTANT: It is JCB policy that if requested, original data images must be made available. Failure to provide original images upon request will result in unavoidable delays in publication. Please ensure that you have access to all original microscopy and blot data images before submitting your revision.

Supplemental information: There are strict limits on the allowable amount of supplemental data. Your manuscript may have up to 5 supplemental figures. Up to 10 supplemental videos or flash animations are allowed. A summary of all supplemental material should appear at the end of the Materials and methods section.

If you choose to resubmit, please include a cover letter addressing the reviewers' comments point by point. Please also highlight all changes in the text of the manuscript.

Regardless of how you choose to proceed, we hope that the comments below will prove constructive as your work progresses. We would be happy to discuss them further once you've had a chance to consider the points raised. You can contact the journal office with any questions, cellbio@rockefeller.edu or call (212) 327-8588.

Thank you for thinking of JCB as an appropriate place to publish your work.

Sincerely,

Monica Bettencourt-Dias, PhD
Monitoring Editor, Journal of Cell Biology

Melina Casadio, PhD
Senior Scientific Editor, Journal of Cell Biology

Reviewer #1 (Comments to the Authors (Required)):

The manuscript from Lee et al. deals with the role of Cep97 and Dyrk1a in the formation of motile cilia in the skin of *Xenopus* embryos. They identify an interaction between these proteins via mass spec and validate and map the interaction domains using co-IP. They go on to show that Dyrk1a phosphorylates Cep97 and that this phosphorylation is important. MO knockdown of Cep97 has ciliogenesis defects that are rescued by WT or a phosphomimetic version of Cep97 but not by a non-phosphorylatable Cep97. Similarly, loss of Dyrk1a causes ciliogenesis defects that are rescued by WT Dyrk1a but not by a version that does not interact with Cep97. They identify an interaction between Cep97 and Plk1 that is disrupted by a loss of phosphorylation and which is important to form a complex with Dyrk1a. They identify the underlying defects ciliogenesis as an inability of centrioles to disengage from the deuterosome and properly dock, a phenotype that is rescued by overexpression of the WT separase but not an inactive separase. The paper is logical and well

written and the claims are mostly justified by the data. I think it adds considerably to the field and provides a useful context dependent function for Cep97 and Dyrk1a in MCCs.

Comments:

1. 1H. The authors provide two exposures which helps with the Dvl2 control, but the even the short exposure does not allow for a clear distinction on bands in the Cep97 lanes. If available it would be nice to replace the short expose with an even shorter exposure.

2. My biggest concern with this paper involves the data in Figure 5 and 6 that utilizes the transgenic Deup1-GFP line. The authors propose that a previous report showed a decrease in Deup1-GFP in control MOs. I find this result confusing and difficult to believe. I have scoured that reference and there are several issues. One is that paper does not show or claim a complete loss of Deup1-GFP but only that they don't see any deuterosome localization after bb docking. If I understand things correctly, that paper used mRNA injections of Duep1-GFP and they explicitly state they use low amounts of expression. At the same time they still show co-localization with centrin at presumptive centrioles. In this paper the authors use the Tub-Deup1 transgenic line which 1.) expresses at a very high levels based on the strong tubulin promoter and 2.) is also seen at both centrioles and deuterosomes (which is not clearly stated and complicates all interpretations). Furthermore Kim et al 2020 do not report a loss of Deup1-GFP in control MO cells using this transgenic line. Yet in the "representative images" there is a complete lack of ANY fluorescence (not just a decrease). If this is true, it is a really odd finding that needs to be addressed before this can be a useful reagent. Also given the localization of both Deup1, Cep97 and Dyrk1a at or near centrioles, a centriole marker should be included to distinguish between deup1 at the centrioles and at the deuterosomes (alternatively use a different deuterosome marker), and this analysis should be done specifically below the apical surface to distinguish between centrioles that are docked (and therefore disengaged) or below the surface and therefore potentially not disengaged. A mosaic experiment would show a clear distinction between Tub-Deup1-GFP + and - control MO (or other MO). The complete lack of any fluorescence is simply hard to digest. Larger areas should be shown as well.

3. Nowhere in the methods did I see it described that the images were taken at the same microscope settings (e.g. laser power). All figures in which there is a comparison (not just quantification) of the amount of fluorescence (especially Figs 5 and 6) need to be done at the same settings and this needs to be explicitly stated.

Minor.

Figure 2A-C. It seems that there is lot of wasted space in this figure. It would help if you could use that space to increase the size of the images as I worry that in journal format they will be even smaller, which might make the data difficult to assess.

As expected, MCCs from control MO-injected embryos displayed decreased fluorescent signal for Deup1-GFP at stage 23 (Revinski et al., 2018), which is in line with a previous report (Figs. 4, C and D)....should be Figs 5 C and D.

Reviewer #2 (Comments to the Authors (Required)):

Specialized multiciliated cells (MCCs) elaborate dense motile cilia that beat in a co-ordinated manner to drive fluid flow over epithelial surfaces. Defects in motile cilia formation or beating lead to

fluid buildup in the brain, increased respiratory tract infections, and infertility. Centrioles reside at the base of each cilium and serve as a template for cilium assembly. In cycling cells, centriole formation is tightly controlled so that a single new procentriole forms adjacent to each of the two parent centrioles. However, MCCs deviate from this strict numerical control to produce hundreds of centrioles to serve as the foundation for building hundreds of motile cilia. How MCCs co-ordinate massive centriole amplification remains unclear.

In this manuscript, Lee et al., proposed that a protein complex comprised of Cep97, Dyrk1a, and Plk1 is required for centriole disengagement and docking during multiciliogenesis. Using in vitro kinase assays and immunoprecipitation experiments, the authors show that the kinase Dyrk1a interacts with Cep97 and phosphorylates it at 4 serine residues. This allows phosphorylated Cep97 to interact with Plk1. Phosphorylation of Cep97 is shown to be required for centriole disengagement and efficient cilia assembly in *Xenopus* MCCs.

This is an interesting and exciting manuscript that provides valuable new insights into how centriole disengagement is regulated in MCCs. The experiments are carefully performed, and the effects of Cep97 or Dyrk1a knockdown are clear and accurately described. However, the mechanism by which Cep97 controls centriole assembly in MCCs requires further clarification before publication in JCB can be recommended.

Major comments

1. The phenotype resulting from the knockdown of Cep97 or Dyrk1a is a little confusing. In Figure 5, the authors show that deuterostomes persist longer in the absence of Cep97 or Dyrk1a and also show centriole disengagement is prevented. However, it remains unclear if centriole disengagement is blocked entirely or merely delayed. A longer time course may help to clear this up. Indeed, the data in Figure 5C indicates more disengaged centrioles at stage 25 compared to stage 23 in the MO conditions. Figure 3A shows there is a reduction in centriole number in cells depleted of Cep97 or Dyrk1a. How this phenotype is related to the block/delay in centriole disengagement remains unclear. On top of this, in figures 2E and 3A, the authors show that there are numerous undocked but disengaged centrioles in the Cep97 or Dyrk1a depleted cells. Is there a connection between the delay in disengagement and failure to dock? Plk1 has been implicated in centriole disengagement and distal appendage assembly. Thus, the delay in centriole disengagement and the inability to dock may be caused by reduced Plk1 activity in Cep97 or Dyrk1a depleted cells. Do the centrioles in these cells contain distal appendages?
2. It is unclear how Plk1 binds to Cep97. Plk1 binds to phosphorylated substrates through its Polo-box domain - is this case for Cep97 binding? Also, the Dyrk1a phosphorylation sites in Cep97 do not resemble known Plk1 docking sites, raising the question of how Plk1 would bind phosphorylated Cep97. These points should be clarified.
3. In figure 6, the role of separase in centriole disengagement from deuterosomes should be further characterized. The role of separase in promoting centriole disengagement in cycling cells is controversial. Furthermore, it has not been clearly established that separase activity is required for centriole disengagement in MCCs. The authors should test if separase knockdown or expression of non-degradable securin blocks centriole disengagement in *Xenopus* MCCs.
4. The localization of Cep97 is not well characterized. The authors should determine if Cep97 is only localized to procentrioles attached to deuterosomes or if it is present on disengaged centrioles. Additionally, is the localization of Cep97 altered in the absence of Plk1 or Dyrk1?
5. The mass spectrometry data identifying Dyrk1a as a Cep97 interacting partner should be shown. Similarly, the mass spectrometry data identifying phosphorylated residues in Cep97 should be shown. It is unclear if Serines 634, 643, 649, and 653 are the only residues found to be phosphorylated in the in vitro kinase assay.

6. To supplement figure 2A-D, the authors should include a quantification of the total percentage of multiciliated cells. It is unclear how the authors identify the MCCs to measure the intensity of acetylated tubulin. If acetylated tubulin levels are low in MCCs, can these cells still be reliably identified?

7. As part of figure 2, the authors should show the expression levels of Cep97 SA and Cep97 SD to show these constructs are expressed at similar levels to endogenous Cep97.

Minor comments

1. The cells in the bottom panel of figure 2A (Cep97 MO + SD) appear to have a different morphology.

2. In Figure 2E, the authors show that the cilia are shorter in the cells depleted of Cep97 and Dyrk1a. The authors don't address this phenotype.

Reviewer #3 (Comments to the Authors (Required)):

The manuscript by Lee et al., provides molecular insights into the mechanism driving centriole disengagement in *Xenopus* larvae skin multiciliated cells. They identify Dyrk1a and Cep97 as major regulator of this process, with Dyrk1a phosphorylating Cep97, which is needed for the action of separase, probably through the recruitment of Plk1. The experiments are nicely done and rigorously analyzed. The paper is clearly written and data are very convincing for most of them. This paper provides novel and significant mechanistic insight into cellular function that will be of interest to a general readership and deserve publication in JCB. Some experiments need however to be done or precised to fully support all the conclusions that are made.

Major points:

-Could the MS data after Cep97 IP be inserted in sup file? At least, the authors should comment on the presence (or absence) of Cdc20b and Spag5 in the immunoprecipitated complex. Cdc20b was proposed to be in complex with Plk1 (so is Cep97) and to be involved in centriole disengagement in the same cellular model (Revinski 2018), it is important that the authors put their findings in this context (in the result as well as discussion sections).

-The authors state that Cep97 or Dyrk1a mutations lead to a decreased number of centrioles. In the case of a disengagement problem, centrioles are very close to each other making it very difficult to count. The decrease in the number of discernible centrioles is probably due to the spatial resolution of the image that does not allow discriminating two closely located centrioles. This could also be the case for the cdc20b paper. I don't think one should conclude on the number of centrioles in this case (as the pictures provided confirm). I would strongly suggest removing these conclusions, as they are not supported by convincing data.

-The authors state that Cep97 is a scaffold between Dyrk1a and Plk1. While a decreased of Plk1 binding to Dyrk1a is observed with Cep97 MO and rescued by MO resistant Cep97 expression (Fig. 1F), the experiment is only conducted once and not quantified. As it is a strong conclusion of the paper, I would suggest repeating the experiment and quantifying it. Their conclusion is strengthened by the fact that the delta HIS mutant of Dyrk1a, which cannot bind to Cep97, fails to bind to Plk1. But one can argue that the HIS domain is required for Dyrk1a to bind to Plk1. Also, is Plk1 retrieved in the MS of IPed Cep97 (figure 1)?

-The colocalization of Cep97 with Deup1 is not very convincing while Plk1 colocalize perfectly with Deup1 in the pictures provided (Fig. 4 and 5). Could the authors provide more pictures? This brings me to the point that the sequence of events from Cep97 to separase is not very clear. The model in Fig. 6E is consequently a bit blurry for this aspect. Is it Cep97, which is known to be recruited at the growing procentrioles that recruit Dyrk1a which recruit Plk1? Could the authors analyze how the localization of each partner is affected by the different interaction mutants to precise the molecular cascade?

-The role of Cep97 in centriole structure and ciliation has to be precised in the context of the literature.

1-In GRP, is the phenotype of Cep97MO comparable to what is described in drosophila (Dobbelaere 2020)?

2-In MCC, is Cep97 only required for centriole disengagement? Meaning, could the authors show that the ciliation and centriole structure are really ok in Cep97MO+separase? The ciliation is quantified by measurement of actub fluorescence intensity, which is not sufficient. At least the authors should show high-resolutive images of basal bodies + cilia in Cep97MO+separase (in z-section for example). Also cilia length should be measured. In Cep97 and Dyrk1a MO, cilia length is strongly decreased (Fig. 2G), which is difficult to attribute to incorrect disengagement only. If a basal body is correctly structured and cilia machinery is ok, then, when the BB docks, the cilium should be ok. Although apical actin could be involved in impaired ciliary length (see next point). Do the author have sufficient TEM sections to show if centrioles in Cep97MO have correct structure? In any way, if Cep97 affect centriole/cilia structure or cilia length, this does not affect the message or interest of the paper. But the results should be analyzed -or at least discussed for the points needing difficult experiments- in the context of what is already known for Cep97.

-It is unclear whether apical actin alteration is due to Cep97/Dyrk1a downregulation or result from the problem of BB docking which fail to organize apical actin network. One way to answer easily would be to stain phalloidin in Cep97MO+separase.

Minor points:

-Fig1b: could the author comment on the presence of 2 bands for Dyrk1a ?

-Fig 1E: the authors state that the deltaC mutant of Dyrk1a binds to Cep97 but compared to the control, the interaction is strongly decreased, this should be mentioned.

-Fig1H: the shift of the Cep97 band is not very visible. Could the authors expose less the upper panel? Since there is a lower panel showing strong exposition for Dvl2 data, there is no need to see Dvl2 in the upper panel.

-Fig1J: I'm not an expert in MS but should the authors provide the raw data allowing them to conclude on the phosphorylation sites?

-page11: the authors say that Deup1 is dissociated from deuterosomes upon centriole maturation. It is not the case, Deup1 is the core component of the deuterosomes. No Deup1, no deuterosome (Zhao 2013, Mercey 2019)

-Fig6E: add separase on the scheme and if possible precise the sequence of events

-Legends: In general, legends need to be proofread, as they do not always describe properly what is shown (examples: Fig1H-J: the associated legend should be developed, Fig2E: what are the structures pointed by the arrows..).

Dear Editor and Reviewers,

We are submitting a revised manuscript (#202102110R) entitled “**Dyrk1a phosphorylation of CEP97 is critical for centriole separation during multiciliogenesis**”. We greatly appreciate the suggestions and comments of the editor and reviewers, and a number of new experiments have been performed to address the concerns that were raised. We feel that in this revised paper, our use of a combination of loss-of-function (endogenous) experiments, replacement experiments (knockdown of endogenous protein and re-expression at carefully titrated levels), and *in vivo* and *ex vivo* assays in *Xenopus* embryos provide mechanistic insight into how Dyrk1a and CEP97 functionally affect ciliogenesis through regulating centriole separation. We are grateful to the reviewers whose suggestions have led to a more thorough assessment of Dyrk1a and CEP97 and their role in this process, and to strengthen the claims of our paper.

The concerns of reviewers and editor have been addressed below:

Reviewer #1:

The manuscript from Lee et al. deals with the role of Cep97 and Dyrk1a in the formation of motile cilia in the skin of Xenopus embryos. They identify an interaction between these proteins via mass spec and validate and map the interaction domains using co-IP. The go onto show that Dyrk1a phosphorylates Cep97 and that this phosphorylation is important. MO knockdown of Cep97 has ciliogenesis defects that are rescued by WT or a phosphomimetic version of Cep97 but not by a non-phosphorylatable Cep97. Similarly, loss of Dyrk1a causes ciliogenesis defects that are rescued by WT Dyrk1a but not by a version that does not interact with Cep97. They identify an interaction between Cep97 and Plk1 that is disrupted by a loss of phosphorylation and which is important to form a complex with Dyrk1a. They identify the underlying defects ciliogenesis as an inability of centrioles to disengage from the deuterosome and properly dock, a phenotype that is rescued by overexpression of the WT separase but not an inactive separase. The paper is logical and well written and the claims are mostly justified by the data. I think it adds considerably to the field and provides a useful context dependent function for Cep97 and Dyrk1a in MCCs.

Comments:

1. 1H. The authors provide two exposures which helps with the Dvl2 control, but the even the short exposure does not allow for a clear distinction on bands in the Cep97 lanes. If available it would be nice to replace the short expose with an even shorter exposure.

As the reviewer suggested, we replaced old blots with new ones for a clear distinction of bands in the Cep97 lanes.

2. My biggest concern with this paper involves the data in Figure 5 and 6 that utilizes the transgenic Deup1-GFP line. The authors propose that a previous report showed a decrease in Deup1-GFP in control MOs. I find this result confusing and difficult to believe. I have scoured that reference and there are several issues. One is that paper does not show or claim a complete loss of Deup1-GFP but only that they don't see any deuterosome localization after bb docking. If I understand things correctly, that paper used mRNA injections of Duep1-GFP and they explicitly state they use low amounts of expression. At the same time they still show co-localization with centrin at presumptive centrioles. In this paper the authors use

the Tub-Deup1 transgenic line which 1.) expresses at a very high levels based on the strong tubulin promoter and 2.) is also seen at both centrioles and deuterosomes (which is not clearly stated and complicates all interpretations). Furthermore Kim et al 2020 do not report a loss of Deup1-GFP in control MO cells using this transgenic line. Yet in the "representative images" there is a complete lack of ANY fluorescence (not just a decrease). If this is true, it is a really odd finding that needs to be addressed before this can be a useful reagent.

Also given the localization of both Deup1, Cep97 and Dyrk1a at or near centrioles, a centriole marker should be included to distinguish between deup1 at the centrioles and at the deuterosomes (alternatively use a different deuterosome marker), and this analysis should be done specifically below the apical surface to distinguish between centrioles that are docked (and therefore disengaged) or below the surface and therefore potentially not disengaged.

A mosaic experiment would show a clear distinction between Tub-Deup1-GFP + and - control MO (or other MO). The complete lack of any fluorescence is simply hard to digest. Larger areas should be shown as well.

The reviewer's comments are well-taken, and to address the concerns regarding the utilization of the transgenic Deup1-GFP line, we injected mRNAs of GFP-ccdc78 (an alternative deuterosome marker; Figs. 5, 6, and 8) or Deup1-GFP for the centriole disengagement experiments (Fig. S5). Regardless, the GFP-Ccdc78 clearly marks deuterosomes as in the previous report (Dehring et al., 2013). Using GFP-Ccdc78, we show that defective centriole disengagement is evidently reproduced in Cep97 and Dyrk1 morphant MCCs, and mutant Cep97 (SA) expression did not rescue the phenotype unlike wild-type expression (Figs. 5, 6 and 8). We included the data using GFP-Ccdc78 in Figs. 5, 6, and 8 and the data using Deup1-GFP in Fig. S5. We note that expression of GFP-Deup1 expression was still weak in controls when compared to the CEP97 and Dyrk1a morphants.

Also, following the reviewer's suggestion, we provide larger area views in Figures 5, 6 and 8.

To obtain a clear localization of Cep97 at deuterosomes and centrioles, we used GFP-ccdc78 as a deuterosome marker and showed Cep97 is located near or at the deuterosome (Fig. 5A). Interestingly, while centrioles disengage from deuterosomes at stage 19, most of the Cep97 signal did not colocalize with free (disengaged) centrioles, but rather colocalized to the centrioles attached to deuterosomes, suggesting that the colocalization of Cep97 and centrioles may be limited to the deuterosome area. We included the data in Fig. 5 A.

3. Nowhere in the methods did I see it described that the images were taken at the same microscope settings (e.g. laser power). All figures in which there is a comparison (not just quantification) of the amount of fluorescence (especially Figs 5 and 6) need to be done at the same settings and this needs to be explicitly stated.

The reviewer's comment is well-taken. All sets of an individual experiments were performed under the same microscope settings, and all images were processed under identical conditions. For example, the experiment shown in Fig. 5 B was conducted at these settings: 488 channel (Gain(master) 850, digital offset 33, digital gain 1.0), 568 channel (Gain(master) 850, digital 45, digital gain 1.3, 647 channel (Gain(master) 700, digital 150, digital gain 1.0). Even though the same doses of mRNAs were injected into embryos at the same developmental stages, the signal intensity may vary somewhat between biological repeats of a sample set. Thus, whenever a new experiment was performed, all

settings for an experiment were re-adjusted and saturation was avoided using “range indicator”. The new setting was consistently applied for all groups of samples of a particular experimental set. We now clarify this in the Materials and methods section.

Minor.

Figure 2A-C. It seems that there is lot of wasted space in this figure. It would help if you could use that space to increase the size of the images as I worry that in journal format they will be even smaller, which might make the data difficult to assess.

As the reviewer suggested, we expanded the size of the images.

As expected, MCCs from control MO-injected embryos displayed decreased fluorescent signal for Deup1-GFP at stage 23 (Revinski et al., 2018), which is in line with a previous report (Figs. 4, C and D)...should be Figs 5 C and D.

In accordance with the reviewer’s suggestion, we have revised these experiments using *ccdc78* as a deuterosome marker and Deup1-GFP mRNA (rather than transgenics) as mentioned above, and the transgenic data is no longer displayed.

Reviewer #2 :

Specialized multiciliated cells (MCCs) elaborate dense motile cilia that beat in a co-ordinated manner to drive fluid flow over epithelial surfaces. Defects in motile cilia formation or beating lead to fluid buildup in the brain, increased respiratory tract infections, and infertility. Centrioles reside at the base of each cilium and serve as a template for cilium assembly. In cycling cells, centriole formation is tightly controlled so that a single new procentriole forms adjacent to each of the two parent centrioles. However, MCCs deviate from this strict numerical control to produce hundreds of centrioles to serve as the foundation for building hundreds of motile cilia. How MCCs co-ordinate massive centriole amplification remains unclear.

*In this manuscript, Lee et al., proposed that a protein complex comprised of Cep97, Dyrk1a, and Plk1 is required for centriole disengagement and docking during multiciliogenesis. Using in vitro kinase assays and immunoprecipitation experiments, the authors show that the kinase Dyrk1a interacts with Cep97 and phosphorylates it at 4 serine residues. This allows phosphorylated Cep97 to interact with Plk1. Phosphorylation of Cep97 is shown to be required for centriole disengagement and efficient cilia assembly in *Xenopus* MCCs.*

This is an interesting and exciting manuscript that provides valuable new insights into how centriole disengagement is regulated in MCCs. The experiments are carefully performed, and the effects of Cep97 or Dyrk1a knockdown are clear and accurately described. However, the mechanism by which Cep97 controls centriole assembly in MCCs requires further clarification before publication in JCB can be recommended.

Major comments

1. The phenotype resulting from the knockdown of Cep97 or Dyrk1a is a little confusing. In Figure 5, the authors show that deuterosomes persist longer in the absence of Cep97 or Dyrk1a and also show

centriole disengagement is prevented. However, it remains unclear if centriole disengagement is blocked entirely or merely delayed. A longer time course may help to clear this up.

The reviewer makes an interesting point, and to address this question, we included a group of stage 27 embryos. Although the percentage of cells having centriole disengagement defects decreased in stage 25 and stage 27 embryos to some extent, over 40 to 50 percent of the Cep97 and Dyrk1a morphant MCCs still harbored centrioles with incomplete disengagement. Also, we attempted to perform the assay using stage 30 and later embryos, however, the GFP-ccdc78 signal in the embryos at the later stages was not strong enough to assess the deuterosome-centriole disengagement. We included the data comparing stage 23, 25, and 27 in Fig. S5 B, C, D.

Indeed, the data in Figure 5C indicates more disengaged centrioles at stage 25 compared to stage 23 in the MO conditions. Figure 3A shows there is a reduction in centriole number in cells depleted of Cep97 or Dyrk1a. How this phenotype is related to the block/delay in centriole disengagement remains unclear.

The reviewer makes an excellent point, as does reviewer #3 who points out the difficulty in accurately determining centriole numbers. We agree that particularly in the case of a morphant disengagement defects, centrioles are in very close proximity to each other - making it challenging to accurately measure the numbers. Thus, we agree with the reviewers and have removed the centriole number data.

On top of this, in figures 2E and 3A, the authors show that there are numerous undocked but disengaged centrioles in the Cep97 or Dyrk1a depleted cells. Is there a connection between the delay in disengagement and failure to dock? Plk1 has been implicated in centriole disengagement and distal appendage assembly. Thus, the delay in centriole disengagement and the inability to dock may be caused by reduced Plk1 activity in Cep97 or Dyrk1a depleted cells. Do the centrioles in these cells contain distal appendages?

The reviewer makes an excellent point. To answer the reviewer's question, we immunostained endogenous Cep164, a distal appendage marker, and centrin as a basal body marker. In the control morphant MCCs, distal appendages were properly formed and all centrioles contained distal appendages. The distal appendage was also formed in Cep97 and Dyrk1a morphant MCCs. Also, the distal appendage and centriole complexes were often aggregated, implying that defective centriole disengagement may result in abnormal centriole spacing. We now present this data in Fig. 3 D.

2. It is unclear how Plk1 binds to Cep97. Plk1 binds to phosphorylated substrates through its Polo-box domain - is this case for Cep97 binding?

The reviewer proffers an interesting question. We performed interaction domain mapping using deletion mutants of Plk1. A mutant lacking the Polo-Box (PB) domains (1 & 2) lost the interaction with Cep97, but the PB domains alone were sufficient to interact with Cep97. Thus, Plk1 binds to phosphorylated Cep97 through the PB domains. We now include the data in Figs. 4 E and F.

Also, the Dyrk1a phosphorylation sites in Cep97 do not resemble known Plk1 docking sites, raising the question of how Plk1 would bind phosphorylated Cep97. These points should be clarified.

The reviewer is correct that the Dyrk1a phosphorylation site in Cep97 does not resemble the core consensus motif for Plk1 docking (S-(pT/pS)-(P/X) (X, any amino acid except Cys) (Elia AE, Cantley LC, 2003). Plk1 can associate with its binding partners in a non-canonical manner. For example, the interaction between Polo and Map205 does not require the priming phosphorylation of Map205 although the Polo-box domain is still required for the interaction (Archambault et al., 2008). Also, although three phosphorylated residues (T40, S48, and S52) of the Matrimony protein are required for a Polo kinase interaction, these phosphorylated residues do not resemble the core consensus motif for Plk1 docking (Bonner et al., 2013).

We now mention in the results section that the site on CEP97 is not a canonical PLK1 docking site.

3. In figure 6, the role of separase in centriole disengagement from deuterosomes should be further characterized. The role of separase in promoting centriole disengagement in cycling cells is controversial. Furthermore, it has not been clearly established that separase activity is required for centriole disengagement in MCCs. The authors should test if separase knockdown or expression of non-degradable securin blocks centriole disengagement in Xenopus MCCs.

The reviewer makes an excellent suggestion and therefore we tested whether Separase is required for centriole disengagement in embryos. First, we tested the impact of Separase knockdown and whether wild type or mutant Separase expression (lacking enzymatic activity) rescues centriole disengagement. Separase knockdown causes centriole disengagement defects in MCCs at stage 23. Re-expressing wild-type Separase partially rescues the defects, but the inactive mutant fails to do so. Moreover, ectopic expression of the mutant Separase causes significant disengagement defects in a dose-dependent manner when compared to the similar amount of wild-type Separase expression. The data suggests a requirement for Separase in centriole disengagement in the Xenopus system. This data is now presented in Fig. 6.

4. The localization of Cep97 is not well characterized. The authors should determine if Cep97 is only localized to procentrioles attached to deuterosomes or if it is present on disengaged centrioles.

The reviewer's points are well-taken. To better characterize Cep97 localization, we utilized GFP-ccdc78 as a deuterosome marker and centrin as a centriole marker, and performed 3D-SIM for better resolution. In the centriole disengagement stage (19), Cep97 localized near or at GFP-ccdc78, and Cep97 colocalized with centrin attached to the deuterosome, but scarcely colocalized with disengaged (free) centrioles. We now include this data in Fig. 5 A.

Additionally, is the localization of Cep97 altered in the absence of Plk1 or Dyrk1?

The knockdown of Dyrk1a did not affect Cep97 localization to the deuterosome, suggesting that this Cep97 localization is Dyrk1a-independent. The severe deleterious effects on embryogenesis caused by Plk1 knockdown precluded us from testing the Cep97 localization in the absence of Plk1. We now present the Dyrk1a independence of CEP97 in Fig. S4 A.

5. The mass spectrometry data identifying Dyrk1a as a Cep97 interacting partner should be shown. Similarly, the mass spectrometry data identifying phosphorylated residues in Cep97 should be shown. It is unclear if Serines 634, 643, 649, and 653 are the only residues found to be phosphorylated in the in vitro kinase assay.

We included both mass spectrometry data identifying Dyrk1a as a Cep97 interaction partner and identifying Cep97 phosphorylation sites in supplemental tables (Table S1 and Table S2). These were the major phosphorylation sites identified.

6. To supplement figure 2A-D, the authors should include a quantification of the total percentage of multiciliated cells. It is unclear how the authors identify the MCCs to measure the intensity of acetylated tubulin. If acetylated tubulin levels are low in MCCs, can these cells still be reliably identified?

As the reviewer suggested, we included a quantification of the total percentage of MCCs in the supplemental data section (Figs. S2 E and F). The knockdown of Cep97 and Dyrk1a did not affect the total percentage of MCCs in a field. The MCCs were easily discerned in morphants by the residual acetylated tubulin.

7. As part of figure 2, the authors should show the expression levels of Cep97 SA and Cep97 SD to show these constructs are expressed at similar levels to endogenous Cep97.

As the reviewer suggested, we included the Western blot data showing the exogenous Cep97 proteins (WT, SA, and SD) are expressed at similar levels (Fig. 2 C). We could not compare the exogenous proteins with endogenous Cep97 because an antibody detecting endogenous *Xenopus* Cep97 is not available. However, we believe it is important that WT CEP97 and the phospho-mimic mutant both rescue ciliogenesis in the CEP97 morphant, while the SA mutant fails to do so even though all are expressed at similar levels.

Minor comments

1. The cells in the bottom panel of figure 2A (Cep97 MO + SD) appear to have a different morphology.

The embryo mounting condition sometimes causes this different morphology of embryo epidermis while taking images on microscope. We replaced it with a more representative image.

2. In Figure 2E, the authors show that the cilia are shorter in the cells depleted of Cep97 and Dyrk1a. The authors don't address this phenotype.

We now present the observation of this phenotype in the Results section (bottom of page 8). "Using scanning electron microscopy (SEM), we observed a marked decrease in the population and length of multicilia in the Cep97 and Dyrk1a morphant MCCs". We do not know precisely why the cilia are severely shortened in those MCCs that display reduced cilia.

Reviewer #3:

*The manuscript by Lee et al., provides molecular insights into the mechanism driving centriole disengagement in *Xenopus* larvae skin multiciliated cells. They identify Dyrk1a and Cep97 as major regulator of this process, with Dyrk1a phosphorylating Cep97, which is needed for the action of separase, probably through the recruitment of Plk1. The experiments are nicely done and rigorously analyzed. The paper is clearly written and data are very convincing for most of them. This paper provides novel and significant mechanistic insight into cellular function that will be of interest to a general readership and*

deserve publication in JCB. Some experiments need however to be done or precised to fully support all the conclusions that are made.

Major points:

-Could the MS data after Cep97 IP be inserted in sup file? At least, the authors should comment on the presence (or absence) of Cdc20b and Spag5 in the immunoprecipitated complex. Cdc20b was proposed to be in complex with Plk1 (so is Cep97) and to be involved in centriole disengagement in the same cellular model (Revinski 2018), it is important that the authors put their findings in this context (in the result as well as discussion sections).

As the reviewer suggested, we included the MS data in the supplemental data (Table S1). Although Cdc20B and Spag5 were not found in the MS data, we tested whether Cep97 forms an immune complex with Cdc20B and Spag5. Using HEK293T cells, the exogenous Cep97 (wild-type and Δ 577-610; corresponding to Δ 541-565 Xenopus Cep97 mutant) was expressed with Cdc20B or Spag5 and co-IPs were performed. The wild-type Cep97 associated with both Cdc20B and Spag5, however, the mutant Δ 577-610 lost the interaction, suggesting that the region 577-610 (aka PLK1 binding site) is required for the complex formation with Cdc20b and Spag5. We included the data in Figs. S3 F and G and discussion section.

-The authors state that Cep97 or Dyrk1a mutations lead to a decreased number of centrioles. In the case of a disengagement problem, centrioles are very close to each other making it very difficult to count. The decrease in the number of discernible centrioles is probably due to the spatial resolution of the image that does not allow discriminating two closely located centrioles. This could also be the case for the cdc20b paper. I don't think one should conclude on the number of centrioles in this case (as the pictures provided confirm). I would strongly suggest removing these conclusions, as they are not supported by convincing data.

The reviewer makes an excellent point about the difficulty in accurately determining centriole numbers. We agree that particularly in the case of a morphant disengagement defects, centrioles are in very close proximity to each other - making it challenging to accurately measure the numbers. Thus, we agree with the reviewer #3 and have removed the centriole number data.

-The authors state that Cep97 is a scaffold between Dyrk1a and Plk1. While a decreased of Plk1 binding to Dyrk1a is observed with Cep97 MO and rescued by MO resistant Cep97 expression (Fig. 1F), the experiment is only conducted once and not quantified. As it is a strong conclusion of the paper, I would suggest repeating the experiment and quantifying it.

We apologize for being remiss in not presenting this data in a clear manner and giving the reviewer the wrong impression about repetitions etc. We now present the quantified data obtained from three independent experiments. The co-precipitated bands were normalized to input and we included the graph in Fig. 4 H.

Their conclusion is strengthened by the fact that the delta HIS mutant of Dyrk1a, which cannot bind to Cep97, fails to bind to Plk1. But one can argue that the HIS domain is required for Dyrk1a to bind to Plk1. Also, is Plk1 retrieved in the MS of IPed Cep97 (figure 1)?

We understand the reviewer's argument but in figure 1F we show the Dyrk1a Δ His mutant fails to interact with CEP97, while in figure 4G we show that loss of endogenous CEP97 reduces the interaction between exogenously expressed Dyrk1a and Plk1. This is why we believe the data suggest that Dyrk1a Δ His affects the interaction with CEP97, indicating it bridges the Dyrk1a/Plk1 interaction. Strengthening this concept, we identified Plk1 in the IP-MS data, and we now include the MS data in Table S1.

-The colocalization of Cep97 with Deup1 is not very convincing while Plk1 colocalize perfectly with Deup1 in the pictures provided (Fig. 4 and 5). Could the authors provide more pictures?

To better characterize Cep97 localization, we utilized GFP-ccdc78 as a deuterosome marker and centrin as a centriole marker, and 3D-SIM was performed for better resolution. In the centriole disengagement stage (19), Cep97 localized near or at GFP-ccdc78, and Cep97 colocalized with centrin attached to the deuterosome, but scarcely colocalized with disengaged (free) centrioles. We now include this data in Fig. 5 A.

This brings me to the point that the sequence of events from Cep97 to separase is not very clear. The model in Fig. 6E is consequently a bit blurry for this aspect. Is it Cep97, which is known to be recruited at the growing procentrioles that recruit Dyrk1a which recruit Plk1? Could the authors analyze how the localization of each partner is affected by the different interaction mutants to precise the molecular cascade?

Although CEP97 needs to be phosphorylated by Dyrk1a in order for Plk1 to bind to CEP97, we attempted to clarify the sequence of events using localization of each protein in the absence of another (via MO injection) in MCCs, but this was uninformative. The knockdown of Dyrk1a did not change the deuterosome localization of GFP-tagged versions of Cep97 and Plk1, and Cep97 knockdown did not affect the Plk1 location in MCCs, suggesting that the localization of Cep97 and Plk1 to deuterosomes may be an independent event. Rather, their interactions may be important for timely Separase activation by modulating access of Plk1-Cdc20B-Spag5 complex to Separase itself or Separase regulating proteins (e.g. Securin) in the centriole-deuterosome. We have included this negative data showing the localization of GFP-tagged Cep97 and Plk1 in Dyrk1a and Cep97 morphant MCCs in Fig. S4.

-The role of Cep97 in centriole structure and ciliation has to be precised in the context of the literature. 1-In GRP, is the phenotype of Cep97MO comparable to what is described in drosophila (Dobbelaere 2020)?

We understand the reviewer's point and attempted to perform TEM for single motile cilia in the GRP multiple times, but this proved to be very elusive due to the low abundance of such cilia and the difficulty of getting a precise orientation in the tissue.

2-In MCC, is Cep97 only required for centriole disengagement? Meaning, could the authors show that the ciliation and centriole structure are really ok in Cep97MO+separase? The ciliation is quantified by measurement of actub fluorescence intensity, which is not sufficient. At least the authors should show high-resolutive images of basal bodies + cilia in Cep97MO+separase (in z-section for example). Also cilia length should be measured.

The reviewer makes several interesting points and suggestions. We will address them in order:

We do not have high-resolution images of basal bodies + cilia to determine whether centriole structures are defective at this time. We did obtain SEM images of Cep97MO + Separase and quantified the cilia length and number which is now presented in Figs 7 D-F.

In Cep97 and Dyrk1a MO, cilia length is strongly decreased (Fig. 2G), which is difficult to attribute to incorrect disengagement only. If a basal body is correctly structured and cilia machinery is ok, then, when the BB docks, the cilium should be ok. Although apical actin could be involved in impaired ciliary length (see next point). Do the author have sufficient TEM sections to show if centrioles in Cep97MO have correct structure?

As suggested, we conducted TEM to observe axonemes in Cep97 and Dyrk1 morphant MCCs. The cross-section images of remaining cilia in the morphant MCCs reveal a typical normal axoneme structure in the morphants.

In any way, if Cep97 affect centriole/cilia structure or cilia length, this does not affect the message or interest of the paper. But the results should be analyzed -or at least discussed for the points needing difficult experiments- in the context of what is already known for Cep97.

-It is unclear whether apical actin alteration is due to Cep97/Dyrk1a downregulation or result from the problem of BB docking which fail to organize apical actin network. One way to answer easily would be to stain phalloidin in Cep97MO+separase.

As suggested, we examined apical actin formation and observed the defect was partially reversed by Separase expression in Cep97 and Dyrk1a morphant MCCs (Fig. 7 A). In addition, in Cep97 and Dyrk1 morphants, we observed defective basal body migration and docking, and distal appendages are present (now presented in Fig. 3). Thus, our findings suggest that in the absence of CEP97 or Dyrk1a, the abnormal apical actin formation and impaired basal body docking is caused by a failure of centriole disengagement, thus, resulting in multiciliation defects in Cep97 and Dyrk1a depleted MCCs.

Minor points:

-Fig1b: could the author comment on the presence of 2 bands for Dyrk1a ?

The reviewer makes a perceptive observation. Dyrk1a often shows multiple bands in immunoblots of cell lines. Similar to our data in Fig. 1b, the Luna group showed double bands of endogenous Dyrk1a in immunoblots using HEK293T lysates (Sergi Aranda, 2008). The shifted bands are likely the result of post-translationally modified Dyrk1a (via phosphorylation) because the upper bands of Dyrk1a disappear after incubation U2-OS cell lysates with alkaline phosphatase (Monica Alvarez, 2007).

-Fig 1E: the authors state that the deltaC mutant of Dyrk1a binds to Cep97 but compared to the control, the interaction is strongly decreased, this should be mentioned.

Band intensity measurement clearly showed that comparing with other mutants, Δ HIS interaction with Cep97 significantly decreased in both directions of pulldown (Dyrk1a-Flag pulldown: WT 1, Δ HIS 0.12, Δ C 0.65; Cep97-HA pulldown: WT 1, Δ HIS 0.02, Δ C 0.1 – each band normalized to input). As reviewer pointed out, Δ C mutant showed significantly reduced interaction with Cep97 in one direction of Co-IP (Cep97-HA pulldown), however, the decreased interaction was not convincing and variable in the other direction of Co-IP (Dyrk1a-Flag pulldown). This result was reproduced, and we

concluded the HIS repeat is the most critical region of Dyrk1 in Dyrk1a-Cep97 binding. This is now mentioned in the Results section.

-Fig1H: the shift of the Cep97 band is not very visible. Could the authors expose less the upper panel? Since there is a lower panel showing strong exposition for Dvl2 data, there is no need to see Dvl2 in the upper panel.

We replaced the blot with one run for a longer time period to more clearly demonstrate the shift in the Cep97 band.

-Fig1J: I'm not an expert in MS but should the authors provide the raw data allowing them to conclude on the phosphorylation sites?

As the reviewer suggested, we now provide the MS data on phosphorylation sites in Table S2.

-page11: the authors say that Deup1 is dissociated from deuterosomes upon centriole maturation. It is not the case, Deup1 is the core component of the deuterosomes. No Deup1, no deuterosome (Zhao 2013, Mercey 2019)

We thank the reviewer for catching our error, and we removed this wording from the Results section. In addition, in response to reviewers' suggestions, we performed immunofluorescence assays using N-terminally tagged Ccdc78 as a deuterosome marker with SIM, rather than just the Deup1 (Figs. 5, 6, and 8).

-Fig6E: add separate on the scheme and if possible precise the sequence of events

As the reviewer suggested, we added Separate on the scheme.

-Legends: In general, legends need to be proofread, as they do not always describe properly what is shown (examples: Fig1H-J: the associated legend should be developed, Fig2E: what are the structures pointed by the arrows...).

According to the reviewer's comment, we revised the legends.

All the reviewers' assessments are much appreciated.
We look forward to a positive assessment of this revised manuscript.

Sincerely,

Ira Daar, Ph.D.

Senior Investigator, Chief
Cancer & Developmental Biology Laboratory
NCI, NIH

September 20, 2021

RE: JCB Manuscript #202102110R

Dr. Ira Daar
National Cancer Institute
Cancer & Developmental Biology Laboratory
Bld. 560 rm. 12-88
Frederick, MD 21702

Dear Dr. Daar:

Thank you for submitting your revised manuscript entitled "Dyrk1a phosphorylation of CEP97 is critical for centriole separation during multiciliogenesis". We are happy to say that the reviewers are now supportive of publication. However, reviewers #1 and #3 still have some major concerns that can be addressed by text edits. Please be sure to address all the remaining concerns raised by reviewers in the final version of the manuscript. Pending these revisions and any revisions necessary to meet our length and other formatting guidelines (see details below), we would be happy to publish the paper in JCB.

A. MANUSCRIPT FORMATTING:

Full guidelines are available on our Instructions for Authors page, <http://jcb.rupress.org/site/misc/ifora.xhtml>. Submission of a paper that exceeds these limits without prior discussion with the journal office will delay scheduling of your manuscript for publication.

- 1) Text limits: Character count for Articles and Tools is < 40,000, not including spaces. Count includes title page, abstract, introduction, results, discussion, and acknowledgments. Count does not include materials and methods, figure legends, references, tables, or supplemental legends.
- 2) Figures limits: Articles and Tools may have up to 10 main text figures.
- 3) Figure formatting:
Molecular weight or nucleic acid size markers must be included on all gel electrophoresis. *** Please, add a molecular weight marker on gel in supplementary figure 1B.

Scale bars must be present on all microscopy images, including inset magnifications. *** Please, add scale bars in inset magnifications in main figure 3D.

*** Also, please avoid pairing red and green for images in main figures 2A, 2D, 3A-B, 3D, 4A, 5A-B, 6A-B, 7A, 8A and supplementary figures 1C, 2A, 2C, 2H, 3A-D, 4A, 5A to ensure legibility for color-blind readers.

4) Statistical analysis:

Error bars on graphic representations of numerical data must be clearly described in the figure legend.

The number of independent data points (n) represented in a graph must be indicated in the legend.

*** Please specify in the figure legends whether n refers to the number of quantified cilia or cells.

For instance, in main figure 1: "(B and E) Quantification of relative acetylated tubulin intensity in A (image n = 40 [cilia or MCCs?] from 20 embryos for each condition) and C (image n = 32 [cilia or MCCs?] from 16 embryos for each condition)." Also, although not required, we recommend using SuperPlots to better represent the individual measures from independent embryos and the corresponding averages in the graphs.

Statistical methods should be explained in full in the materials and methods.

For figures presenting pooled data the statistical measure should be defined in the figure legends.

Please also be sure to indicate the statistical tests used in each of your experiments (both in the figure legend itself and in a separate methods section) as well as the parameters of the test (for example, if you ran a t-test, please indicate if it was one- or two-sided, etc.). *** Also, since you used parametric tests, please indicate if the data distribution was tested for normality (and if so, how). If not, you must state something to the effect that "Data distribution was assumed to be normal but this was not formally tested."

5) Abstract and title:

The abstract should be no longer than 160 words and should communicate the significance of the paper for a general audience.

The title should be less than 100 characters including spaces. Make the title concise but accessible to a general readership. *** While your title looks fine, we suggest "CEP97 phosphorylation by Dyrk1a is critical for centriole separation during multiciliogenesis" as we think it is clearer to the readers.

6) Materials and methods: Should be comprehensive and not simply reference a previous publication for details on how an experiment was performed. Please provide full descriptions (at least in brief) in the text for readers who may not have access to referenced manuscripts. The text should not refer to methods "...as previously described."

7) Please be sure to provide the sequences for all of your primers/oligos and RNAi constructs in the materials and methods. You must also indicate in the methods the source, species, and catalog numbers (where appropriate) for all of your antibodies.

8) Microscope image acquisition: The following information must be provided about the acquisition and processing of images:

- a. Make and model of microscope
- b. Type, magnification, and numerical aperture of the objective lenses
- c. Temperature
- d. imaging medium
- e. Fluorochromes
- f. Camera make and model

g. Acquisition software

h. Any software used for image processing subsequent to data acquisition. Please include details and types of operations involved (e.g., type of deconvolution, 3D reconstitutions, surface or volume rendering, gamma adjustments, etc.).

10) Supplemental materials:

There are strict limits on the allowable amount of supplemental data. Articles/Tools may have up to 5 supplemental figures. *** At the moment, you currently have 8 such items (5 figures and 3 tables). While we will be able to give you a bit more space in this case, we will still need for you to reduce the count a bit. Specifically, you could combine tables S1 and S2 into a single supplementary table with two 'panels' (A and B). Please be sure to correct the callouts in the text to reflect this change. In addition, since you only currently have 8 main display items (8 figures), you could also consider moving one or more of the supp figures to the main text.

*** Please also note that tables, like figures, should be provided as individual, editable files.

A summary of all supplemental material should appear at the end of the Materials and methods section.

11) eTOC summary:

A ~40-50 word summary that describes the context and significance of the findings for a general readership should be included on the title page.

*** The statement should be written in the present tense and refer to the work in the third person. It should begin with "First author name(s) et al..." to match our preferred style.

13) A separate author contribution section is required following the Acknowledgments in all research manuscripts. All authors should be mentioned and designated by their first and middle initials and full surnames. We encourage use of the CRediT nomenclature (<https://casrai.org/credit/>).

14) ORCID IDs: ORCID IDs are unique identifiers allowing researchers to create a record of their various scholarly contributions in a single place. At resubmission of your final files, please consider providing an ORCID ID for as many contributing authors as possible.

15) Materials and data sharing: All datasets included in the manuscript must be available from the date of online publication, and the source code for all custom computational methods, apart from commercial software programs, must be made available either in a publicly available database or as supplemental materials hosted on the journal website. Numerous resources exist for data storage and sharing (see Data Deposition: <https://rupress.org/jcb/pages/data-deposition>), and you should choose the most appropriate venue based on your data type and/or community standard. If no

appropriate specific database exists, please deposit your data to an appropriate publicly available database.

FINAL FILES:

In order to accept and schedule your paper, we need you to upload the following materials to eJP. If you have any questions about the online submission of your final materials, please contact JCB's Supervising Manuscript Coordinator, Lindsey Hollander (lhollander@rockefeller.edu).

- 1) Electronic version of the text: An editable version of the final text is needed for copyediting (no PDFs).
- 2) High-resolution figure and video files: Individual high-resolution, editable figure files must be provided for each figure. Acceptable figure file formats are .eps, .ai, .psd, and .tif. JCB cannot accept PowerPoint files. All images must be at least 300 dpi for color, 600 dpi for greyscale and 1,200 dpi for line art. Videos must be supplied as QuickTime files.
- 3) It is JCB policy that if requested, original data images must be made available to the editors. Please ensure that you have access to all original data images prior to final submission.
- 4) Cover images: If you have any striking images related to this story, we would be happy to consider them for inclusion on the cover or table of contents. Images should be uploaded as .tif or .eps files and must be at least 300 dpi resolution.

****The license to publish form must be signed before your manuscript can be sent to production. A link to the electronic license to publish form will be sent to the corresponding author only. Please take a moment to check your funder requirements before choosing the appropriate license.****

You can contact me or the scientific editor listed below at the journal office with any questions, jcellbiol@rockefeller.edu.

Thank you for this interesting contribution, I look forward to publishing your paper in The Journal of Cell Biology.

Sincerely,

Monica Bettencourt-Dias
Monitoring Editor
Journal of Cell Biology

Lucia Morgado-Palacin, PhD
Scientific Editor
Journal of Cell Biology

Reviewer #1 (Comments to the Authors (Required)):

This revised manuscript represents a serious improvement on an already strong manuscript. The authors have dealt with all of my concerns and I think the changes dramatically strengthen the claims. I have two very minor points. First, I think the references used for the link between actin, basal body and migration and docking are good (Antoniades et al., 2014; Werner et al., 2011) but I would suggest also including Kulkarni et al 2018 Dev Cell paper as I think that is also quite relevant. Finally in the discussion (and intro) the authors mention the important fact that Dyrk1a is located in the critical trisomy region of Chr21. While not the focus of this paper they suggest that this could be important for ciliogenic defects in Downs patients. While I think it is fine to mention, it feels inappropriate to ignore (although properly reference) the Galati paper that has pretty solid evidence that PCNT underlies the ciliogenic defects. The way it is presented one would think that dyrk1a is likely to be critical for Downs and I just don't know if that is accurate. There is no evidence in this paper that a trisomic increase in Dyrk1a would be problematic. Perhaps just mentioning that in addition to PCNT, Dyrk1a could contribute to Downs cilia issues would be more appropriate.

Reviewer #2 (Comments to the Authors (Required)):

The authors have addressed all my concerns. I support publication in JCB.

Reviewer #3 (Comments to the Authors (Required)):

The authors addressed convincingly most of my comments. I still have a concern regarding the role of Dyrk1a and Cep97 in centriole maturation as I explain below. I know this can be difficult to address in xenopus as the early steps of centriole amplification are difficult to study. I would suggest to cover it as a hypothesis in the text. I still think this paper provides novel and significant mechanistic insight into cellular function that will be of interest to a general readership and deserve publication in JCB.

Major Comments

1. The staining of Cep164 in Cep97MO, Cep97MO+SA and Dyrk1aMO are clearly different from the WT and Cep97MO+WT. Cep164 seems to be recruited in only some part of the centriole circumference and not all around it. It does not seem to be just a problem of centriole aggregation or tilting. This should definitely be noticed as Plk1 is involved in centriole maturation and it can be an additional explanation for the lack of cilia.

2. Because of this and other reasons I will address in the following, I would not exclude the possibility that Dyrk1a and Cep97 are also involved in centriole maturation. Which would be interesting and logical since Plk1 is also involved in centriole maturation in both cycling cells and MCC. In fact :

- Cep97 is present early in migrating MCC (for Dyrk1a we don't know)
- In the different MO conditions of Fig5 and 6, ccdc78 and centrin staining show aggregations of ccdc78 and centrin where no centrioles can be resolved. This can be due to centriole proximity but

also to a block at earlier step of centriole amplification than disengagement.

- Separase convincingly rescue the disengagement phenotype but the new centrioles may not be fully/properly matured. TEM of figure 2 with few pictures and centrioles outside the section plane, and centrin fluorescence staining are not sufficient to address this issue. Altered Cep164 staining support this hypothesis.
- In the separase rescue, while a clearly better ciliation is observed suggesting that a significant part of the centrioles are docked, the length of cilia is altered. One could argue that centriole maturation is to be blamed. In additions, AcTub staining does not reflect properly the number of cilia wich seem to be decreased on the SEM pictures.

Minor comments

1. In fig. S3B and C, I think the first 3 columns of the IP are without cdc20b (for B) or SPAG5 (for C) presence. And the 4th column is in presence of Cdc20b or SPAG5. Or I don't understand the results.
2. Regarding the role of Plk1 in mitotic progression, the authors should remove the Revinski ref since it is in MCC. And for the role of Plk1 in MCC, the authors should add Al Jord 2017. In the Al Jord paper, in ependymal cells, Plk1 is shown to be involved not only in centriole disengagement but also in centriole maturation, like in cycling cells.
3. Quantifications of the number of cilia with SEM in Fig 2 and 8 should be removed or done differently. On the WT picture it is not possible to count the number of cilia. One could make a binning and count the number of cells with more than 10 cilia for example . This would avoid the main bias existing in cells that have a high number of cilia on top of each others.

Typo errors

Intro:

-typo for the word "accessory"

Second paragraph:

-"in in vivo model systems"; a "in" is lacking

Text

Precise the following sentences as some part are either vague or some logical links are missing/unclear:

Intro:

« Centrosomal protein 97 (CEP97) was originally found to suppress primary cilia formation in collaboration with CP110 by capping the mother centriole (Spektor et al., 2007), and the removal of CEP97 and CP110 from the mother centriole is a prerequisite for primary cilia formation (Huang et al., 2018; Nagai et al., 2018). However, recent studies demonstrate CP110 is required in both primary and multiciliogenesis in vivo in different species (Walentek et al., 2016; Yadav et al., 2016), and CEP97 is reported to also be essential for cilia formation in Drosophila by modulating centriole stability (Dobbelaere et al., 2020). While CP110 collaborates with CEP97 to modulate ciliogenesis in cultured cells, it has been suggested CP110 may regulate ciliogenesis independent of CEP97 in vivo model systems (Dobbelaere et al., 2020; Galletta et al., 2016; Walentek et al., 2016) »

Results

« Although immunofluorescence assays using a confocal microscope showed CEP97-GFP localized near the basal bodies (Centrin4-RFP) in MCCs at developmental stage 23 to some extent, CEP97-GFP was rather broadly distributed over the surface of MCCs (Fig. S3 A), which is consistent with the previous report (Walentek et al., 2016). Dyrk1a GFP showed clear localization at or near Centrin4-RFP in the basal bodies at this stage (Fig. S3 B). In contrast, the immature migrating MCCs (stage 19) displayed CEP97-GFP localization near or at the centrioles (Centrin4-RFP) (Fig. S3, C), suggesting a potential role of CEP97 in the centriole production and/or maturation in MCC progenitors ».

In the figure we see Cep97 at the centrioles in both stages. And at later stage it is also localized in the cytoplasm. Regarding Dyrk1a, it is localized at the centrioles and no cytoplasmic staining is observed.

Dear Drs. Bettencourt-Dias and Morgado-Palacin,

We have uploaded a revised final version of manuscript (#202102110R) now entitled “**CEP97 phosphorylation by Dyrk1a is critical for centriole separation during multiciliogenesis**”. We greatly appreciate the suggestions and comments of the editors and reviewers for the text changes and figure modifications.

We have also included a version of the manuscript with highlighted changes from the previous version for ease of assessment.

The point-by-point modifications are listed below.

Again, thank you for all your help in making this a wonderful study for the *Journal of Cell Biology*.

Sincerely,

Ira Daar, Ph.D.
Senior Investigator, Chief
Cancer & Developmental Biology Laboratory
NCI, NIH

Editor's Comments:

A. MANUSCRIPT FORMATTING:

Full guidelines are available on our Instructions for Authors page, <http://jcb.rupress.org/site/misc/ifora.xhtml>. Submission of a paper that exceeds these limits without prior discussion with the journal office will delay scheduling of your manuscript for publication.

1) Text limits: Character count for Articles and Tools is < 40,000, not including spaces. Count includes title page, abstract, introduction, results, discussion, and acknowledgments. Count does not include materials and methods, figure legends, references, tables, or supplemental legends.

Total character count – 30,110

2) Figures limits: Articles and Tools may have up to 10 main text figures.

3) Figure formatting:

Molecular weight or nucleic acid size markers must be included on all gel electrophoresis. *** Please, add a molecular weight marker on gel in supplementary figure 1B.

◇ **We added a molecular weight marker**

Scale bars must be present on all microscopy images, including inset magnifications. *** Please, add scale bars in inset magnifications in main figure 3D.

◇ **We added scale bars**

*** Also, please avoid pairing red and green for images in main figures 2A, 2D, 3A-B, 3D, 4A, 5A-B, 6A-B, 7A, 8A and supplementary figures 1C, 2A, 2C, 2H, 3A-D, 4A, 5A to ensure legibility for color-blind readers.

◇ **We changed colors**

4) Statistical analysis:

Error bars on graphic representations of numerical data must be clearly described in the figure legend.

The number of independent data points (n) represented in a graph must be indicated in the legend. *** Please specify in the figure legends whether n refers to the number of quantified cilia or cells. For instance, in main figure 1: "(B and E) Quantification of relative acetylated tubulin intensity in A (image n = 40 [cilia or MCCs?] from 20 embryos for each condition) and C (image n = 32 [cilia or MCCs?] from 16 embryos for each condition)." Also, although not required, we recommend using SuperPlots to better represent the individual measures from independent embryos and the corresponding averages in the graphs.

◇ **We delineated the specifics in the figure legends.**

Statistical methods should be explained in full in the materials and methods.

For figures presenting pooled data the statistical measure should be defined in the figure legends.

Please also be sure to indicate the statistical tests used in each of your experiments (both in the figure legend itself and in a separate methods section) as well as the parameters of the test (for example, if you ran a t-test, please indicate if it was one- or two-sided, etc.). *** Also, since you used parametric tests, please indicate if the data distribution was tested for normality (and if so, how). If not, you must state something to the effect that "Data distribution was assumed to be normal but this was not formally tested."

◇ **We now state that "Some parametric tests were performed without the normality test because data distribution was assumed to be normal" in the Material and methods section.**

5) Abstract and title:

The abstract should be no longer than 160 words and should communicate the significance of the paper for a general audience.

The title should be less than 100 characters including spaces. Make the title concise but accessible to a general readership. *** While your title looks fine, we suggest "CEP97 phosphorylation by Dyrk1a is critical for centriole separation during multiciliogenesis" as we think it is clearer to the readers.

◇ **We changed to the suggested title.**

6) Materials and methods: Should be comprehensive and not simply reference a previous publication for details on how an experiment was performed. Please provide full descriptions (at least in brief) in the

text for readers who may not have access to referenced manuscripts. The text should not refer to methods "...as previously described."

7) Please be sure to provide the sequences for all of your primers/oligos and RNAi constructs in the materials and methods. You must also indicate in the methods the source, species, and catalog numbers (where appropriate) for all of your antibodies.

8) Microscope image acquisition: The following information must be provided about the acquisition and processing of images:

- a. Make and model of microscope
- b. Type, magnification, and numerical aperture of the objective lenses
- c. Temperature
- d. imaging medium
- e. Fluorochromes
- f. Camera make and model
- g. Acquisition software
- h. Any software used for image processing subsequent to data acquisition. Please include details and types of operations involved (e.g., type of deconvolution, 3D reconstitutions, surface or volume rendering, gamma adjustments, etc.).

10) Supplemental materials:

There are strict limits on the allowable amount of supplemental data. Articles/Tools may have up to 5 supplemental figures. *** At the moment, you currently have 8 such items (5 figures and 3 tables). While we will be able to give you a bit more space in this case, we will still need for you to reduce the count a bit. Specifically, you could combine tables S1 and S2 into a single supplementary table with two 'panels' (A and B). Please be sure to correct the callouts in the text to reflect this change. In addition, since you only currently have 8 main display items (8 figures), you could also consider moving one or more of the supp figures to the main text.

◇ Pursuant to Lindsey's e-mail, we did not change the format.

*** Please also note that tables, like figures, should be provided as individual, editable files.

◇ We provide the tables as excel documents.

A summary of all supplemental material should appear at the end of the Materials and methods section.

11) eTOC summary:

A ~40-50 word summary that describes the context and significance of the findings for a general readership should be included on the title page.

*** The statement should be written in the present tense and refer to the work in the third person. It should begin with "First author name(s) et al..." to match our preferred style.

◇ Fixed as suggested

Reviewer #1:

This revised manuscript represents a serious improvement on an already strong manuscript. The authors have dealt with all of my concerns and I think the changes dramatically strengthen the claims. I have two very minor points. First, I think the references used for the link between actin, basal body and migration and docking are good (Antoniades et al., 2014; Werner et al., 2011) but I would suggest also including Kulkarni et al 2018 Dev Cell paper as I think that is also quite relevant. Finally in the discussion (and intro) the authors mention the important fact that Dyrk1a is located in the critical trisomy region of Chr21. While not the focus of this paper they suggest that this could be important for ciliogenic defects in Downs patients. While I think it is fine to mention, it feels inappropriate to ignore (although properly reference) the Galati paper that has pretty solid evidence that PCNT underlies the ciliogenic defects. The way it is presented one would think that dyrk1a is likely to be critical for Downs and I just don't know if that is accurate. There is no evidence in this paper that a trisomic increase in Dyrk1a would be problematic. Perhaps just mentioning that in addition to PCNT, Dyrk1a could contribute to Downs cilia issues would be more appropriate.

We now include the Kulkarni reference on page 8 of the manuscript.

We now briefly describe the results from the Galati paper and mention its possible contribution to issues of ciliogenesis in Downs syndrome. Page 16 of the manuscript.

Reviewer #3:

The authors addressed convincingly most of my comments. I still have a concern regarding the role of Dyrk1a and Cep97 in centriole maturation as I explain below. I know this can be difficult to address in xenopus as the early steps of centriole amplification are difficult to study. I would suggest to cover it as a hypothesis in the text. I still think this paper provides novel and significant mechanistic insight into cellular function that will be of interest to a general readership and deserve publication in JCB.

Major Comments

1. The staining of Cep164 in Cep97MO, Cep97MO+SA and Dyrk1aMO are clearly different from the WT and Cep97MO+WT. Cep164 seems to be recruited in only some part of the centriole circumference and not all around it. It does not seem to be just a problem of centriole aggregation or tilting. This should definitely be noticed as Plk1 is involved in centriole maturation and it can be an additional explanation for the lack of cilia.

2. Because of this and other reasons I will address in the following, I would not exclude the possibility that Dyrk1a and Cep97 are also involved in centriole maturation. Which would be interesting and logical since Plk1 is also involved in centriole maturation in both cycling cells and MCC. In fact :

- Cep97 is present early in migrating MCC (for Dyrk1a we don't know)
- In the different MO conditions of Fig5 and 6, ccdc78 and centrin staining show aggregations of ccdc78 and centrin where no centrioles can be resolved. This can be due to centriole proximity but also to a block at earlier step of centriole amplification than disengagement.
- Separase convincingly rescue the disengagement phenotype but the new centrioles may not be fully/properly matured. TEM of figure 2 with few pictures and centrioles outside the section plane, and

centrin fluorescence staining are not sufficient to address this issue. Altered Cep164 staining support this hypothesis.

- In the separate rescue, while a clearly better ciliation is observed suggesting that a significant part of the centrioles are docked, the length of cilia is altered. One could argue that centriole maturation is to be blamed. In addition, AcTub staining does not reflect properly the number of cilia which seem to be decreased on the SEM pictures.

Although the possible centriole maturation defect mentioned by the reviewer would be a downstream effect of the centriole separation defect we are reporting, the reviewer's comments are well taken. Thus, we now more clearly indicate in the results section that the distal appendages, while present, are not completely normal in the Cep97 and Dyrk1a morphants – page 9. In addition, we present in the discussion section that we cannot exclude the possibility that Dyrk1a and CEP97 are also involved in part, in centriole maturation – page 16.

Minor comments

1. In fig. S3B and C, I think the first 3 columns of the IP are without cdc20b (for B) or SPAG5 (for C) presence. And the 4th column is in presence of Cdc20b or SPAG5. Or I don't understand the results.

◇ **We fixed this error.**

2. Regarding the role of Plk1 in mitotic progression, the authors should remove the Revinski ref since it is in MCC. And for the role of Plk1 in MCC, the authors should add Al Jord 2017. In the Al Jord paper, in ependymal cells, Plk1 is shown to be involved not only in centriole disengagement but also in centriole maturation, like in cycling cells.

◇ **As the reviewer suggested, we removed the Revinski reference and added the Al Jord paper reference.**

3. Quantifications of the number of cilia with SEM in Fig 2 and 8 should be removed or done differently. On the WT picture it is not possible to count the number of cilia. One could make a binning and count the number of cells with more than 10 cilia for example. This would avoid the main bias existing in cells that have a high number of cilia on top of each others.

◇ **We understand the reviewer's argument that some percentage of cilia in an SEM will be hidden by overlap from other cilia, making it difficult to obtain a completely accurate number. Therefore, as suggested, we removed the quantification data of the SEMs in Figures 2H and 7F, since the same information can be gleaned from the acetylated tubulin quantifications in these figures.**

Typo errors

Intro:

-typo for the word "accessory"

Second paragraph:

-"in in vivo model systems"; a "in" is lacking

◇ **We have fixed the typos.**